

# Atmospheric boundary layer modeling based on mesoscale tendencies and data assimilation at microscale

Javier Sanz Rodrigo[1], Matthew Churchfield[2], Branko Kosovic[3]

[1]Wind Energy department, National Renewable Energy Centre (CENER), Sarriguren, 31621, Spain

[2]National Wind Technology Center, National Renewable Energy Laboratory (NREL), Golden, 80401, CO, U.S.A.

[3]Research Applications Laboratory, National Center for Atmospheric Research (NCAR), Boulder, 80307, CO, U.S.A.

*Correspondence to*: Javier Sanz Rodrigo (jsrodrigo@cener.com)

**Abstract.** The GEWEX Atmospheric Boundary-Layer Studies (GABLS) 1, 2 and 3 are used to develop a methodology for
the design and testing of Reynolds-Averaged Navier Stokes (RANS) atmospheric boundary-layer (ABL) models for wind
energy applications. The first two GABLS are based on idealized boundary conditions and are suitable for verification
purposes by comparing with results from higher-fidelity models based on large-eddy simulation. Results from three single-
column RANS models, of 1st, 1.5th and 2nd turbulence closure order, show high consistency in predicting the mean flow.
The third GABLS case is suitable for the study of these ABL models under realistic forcing such that validation versus
observations at the Cabauw met tower is possible. The case consists on a diurnal cycle that leads to a nocturnal low-level jet
and addresses fundamental questions related to the definition of the large-scale forcing, the interaction of the ABL with the
surface and the evaluation of model results with observations. The simulations are evaluated in terms of surface-layer fluxes
and wind energy quantities of interest: rotor equivalent wind speed, hub-height wind direction, wind speed shear and wind
direction veer. The characterization of mesoscale forcing is based on spatial and temporal-averaged momentum budget terms
from WRF simulations. These mesoscale tendencies are used to drive single-column models, that were verified previously in
the first two GABLS cases, to first demonstrate that they can produce similar wind profile characteristics as the WRF
simulations even though the more simplified physics. The added value of incorporating different forcing mechanisms in
microscale models is quantified by systematically removing forcing terms in the momentum and heat equations. This
mesoscale-to-microscale modelling approach is affected, to a large extent, by the input uncertainties of the mesoscale
tendencies. Deviations from the profile observations are reduced by introducing observational nudging based on
measurements that are typically available from wind energy campaigns. This allows to discuss the added value of using
remote sensing instruments versus tower measurements in the assessment of wind profiles for tall wind turbines reaching
heights of 200 m.



## 1 Introduction

Wind energy flow models are progressively incorporating more realistic atmospheric physics in order to improve the simulation capacity of wind turbine and wind farm design tools. Wind resource assessment and wind turbine site suitability tools, dealing with the microscale flow around and within a wind farm, have been traditionally based on site measurements

and flow models relaying on Monin-Obukhov surface-layer theory (MOST) that assume steady-state and neutral atmospheric conditions (Monin and Obukhov, 1954). At larger scales, the long-term wind climatology is typically determined from a combination of historical measurements and simulations from mesoscale meteorological models at a horizontal resolution of a few kilometers. The transition from mesoscale to microscale to come up with a unified model-chain is the main challenge at stake for the next generation of wind assessment tools (Sanz Rodrigo et al., 2016). In order to make this possible,

microscale models have to extend their range to simulate the entire atmospheric boundary layer (ABL) and include relevant physics like Coriolis as well as realistic large-scale forcing and appropriate turbulent scaling, dependent on thermal stratification, from the surface layer to the free atmosphere. The dynamics of these forcings determine the interplay between the wind climatology, relevant for the assessment of the wind resource, and the wind conditions relevant for wind turbine siting.

The design of ABL models for wind energy requires a systematic approach of verification and validation in order to demonstrate consistency of the computational code with the conceptual physical model and quantify deviations with respect to the real world (Sanz Rodrigo et al., 2016). The verification process is carried out using idealized test cases where the solution is known from theory or from a higher-fidelity model (code-to-code comparison). Sensitivity analysis in idealized conditions also helps determining which are the main drivers of the model, that directly affect the quantities of interest (QoI),

and anticipate their main sources of uncertainty. Validation, in the other hand, deals with code-to-observations comparison to quantify the accuracy of the model at representing the real world in terms of the application of interest. From the wind energy perspective, the quantities of interest are the wind conditions that are directly related to the production of energy and the design characteristics of wind turbines.

The GEWEX Atmospheric Boundary Layer Studies (GABLS) have been developed by the atmospheric boundary-layer

community to benchmark single-column models, used by meteorological models to parameterize the ABL (Holtslag et al., 2013). While the cases are all based on observations of the ABL in relatively stationary and horizontally homogeneous conditions, it is notoriously difficult to define validation cases due to the interplay of a large number of physical processes that can modify these relatively simple conditions. Hence, the first two GABLS benchmarks used idealized conditions in order to analyze the turbulent structure of the ABL without the influence on the variability of the external large-scale

forcing. GABLS1 simulated a quasi-steady stable boundary layer resulting from 9 hours of uniform surface cooling (Cuxart et al., 2006). GABLS2 simulated a diurnal cycle, still with uniform geostrophic forcing, by simplifying measurements from





the CASES-99 experiment in Kansas (Svensson et al., 2011). Under these idealized forcing, large-eddy simulation (LES) models have shown high consistency at predicting the ABL behavior (Beare et al., 2006). Therefore, they have been used to verify reduced-order models based on Reynolds-Averaged Navier Stokes (RANS) turbulence modeling. Hence, GABLS 1 and 2 are suitable verification cases to demonstrate the simulation capacity of ABL models at incorporating thermal
stratification in turbulence modeling under uniform large-scale forcing and using prescribed surface boundary conditions.

GABLS1 showed that many boundary-layer parameterizations tend to overestimate the turbulent mixing in stable conditions leading to a too deep boundary-layer compared to LES simulations (Cuxart et al., 2006). GABLS2 showed the difficulties of comparing observations with simulations under idealized forcing and prescribed surface temperature. Holtslag et al. (2007) showed that during stable conditions there is strong coupling between the geostrophic wind speed and the surface
temperature. Hence, prescribing the surface temperature inhibits the interaction of the boundary-layer with the surface which, for instance, resulted in large differences in the 2-m temperature predicted by the models.

The challenges of the first two GABLS exercises inspired the set-up of GABLS3, which deals with a real diurnal case with a strong nocturnal low-level jet (LLJ) at the Cabauw met tower in the Netherlands (Bass et al., 2009; Bosveld et al., 2014a). Here, large-scale forcing is not constant throughout the diurnal cycle but depends on time and height. Instead of prescribing
the surface temperature, models are allowed to make use of their land-surface schemes in order to include the dependencies between the ABL and the land-surface models. The large-scale forcing is prescribed based on piece-wise linear approximations of the real forcing, derived from simulations with the RACMO mesoscale model and adjusted to match the observed surface geostrophic wind and the wind speed at 200 m. These approximations are introduced to limit the impact of the uncertainties associated to mesoscale geostrophic and advection forcing.

Based on the GABLS benchmark series, the challenges of stable boundary layers and diurnal cycles are reviewed by Hotlslag et al. (2013), notably: the relation between enhanced mixing in operational weather models performance, investigate the role of land-surface heterogeneity in the coupling with the atmosphere, develop LES models with interactive land-surface schemes, create a climatology of boundary-layer parameters (stability classes, boundary-layer depth, and surface fluxes) and develop parameterizations for the very stable boundary layer when turbulence is not the dominant driver.
These challenges are also shared by wind energy applications. Therefore, it is relevant to study the GABLS3 case within the wind energy context as a validation case with focus on rotor-based quantities of interest.

Revisiting GABLS3 for wind energy also means adopting a more pragmatic approach when it comes to adding physical complexity. In the context of developing a mesoscale-to-microscale model it is important to identify which are the first-order physics that need to be incorporated to improve performance against current practices in the wind industry. For instance,
adding thermal effects on turbulence modeling is important against the traditional hypothesis of neutral stratification while the effects of humidity may be initially neglected.





Reducing model-chain uncertainties by using onsite observations is also particularly appealing for wind energy since it is standard practice to count with profile measurements at the site. Since these measurements are typically affected by site effects, we propose to introduce corrections at microscale level based on profile nudging. Hence, contrary to the original GABLS3 set-up, we allow the mesoscale forcing to retain its uncertainties, for the sake of a more generalized mesoscale-to-

microscale methodology, and then relax the microscale model simulation towards the profile observations to correct the hour-to-hour bias. This is also a more natural way of dealing with the wind energy model-chain using an asynchronous coupling methodology where: 1) a database of input forcings is generated offline by a mesoscale model (in the context of a regional wind atlas for instance); 2) site effects are simulated by a microscale ABL model forced by these mesoscale inputs and introducing a high resolution topographic model; and 3) deviations of the model with respect to a reference

observational site are corrected to remove the bias generated throughout the downscaling process. It is important to note that strict validation shall not include site observations to be able to quantify the impact of the limited knowledge of the model. The final data assimilation step allows to calibrate the model to reduce the bias and provide a more accurate wind assessment in the application context. Quantifying the correction introduced by the nudging terms in the modeling equations and their relative weight with respect to the other terms can also be used to assess the limitations of the model.

The methodology used by Bosveld et al. (2014a) to characterize large-scale forcing from mesoscale simulations will be adopted here using simulations from the Weather Research and Forecasting (WRF) model. At microscale, we use a single-column model with three RANS turbulence closure schemes of 1st, 1.5th and 2nd order. This model-chain was also used by Bass et al. (2009) to design the GABLS3 case and perform a sensitivity analysis of various SCM settings. Following a similar philosophy, we evaluate the impact of different mesoscale forcing terms and data assimilation strategies on wind

energy quantities of interest.

## 2 Models

We follow the same modelling approach used by Bass et al. (2010) to define a microscale atmospheric boundary layer model driven by realistic mesoscale forcing. This meso-micro methodology allows to couple the models offline, facilitating the generalization of the downscaling methodology to any combination of mesoscale and microscale models working

asynchronously.

The RANS equations in Cartesian coordinates $(x, y, z)$ for the horizontal wind components $U$ and $V$ are:





$$\frac{1}{f_c}\frac{\partial U}{\partial t} = -\frac{1}{f_c}\left(U\frac{\partial U}{\partial x} + V\frac{\partial U}{\partial y} + W\frac{\partial U}{\partial z}\right) + V - V_g - \frac{1}{f_c}\frac{\partial uw}{\partial z}$$
$$\frac{1}{f_c}\frac{\partial V}{\partial t} = -\frac{1}{f_c}\left(U\frac{\partial V}{\partial x} + V\frac{\partial V}{\partial y} + W\frac{\partial V}{\partial z}\right) - U + U_g - \frac{1}{f_c}\frac{\partial vw}{\partial z}$$ 
(1)

where $f_c$ is the Coriolis parameter, $W$ is the vertical wind component, $U_g$ and $V_g$ are the components of the geostrophic wind and $uw$ and $vw$ are the kinematic horizontal turbulent fluxes for momentum based on the fluctuations about the mean velocity components $u$, $v$, and $w$. For convenience, all the components of the RANS equations have been divided by $f_c$ to

define the equations as the balance of different wind speed vectors:

$$U_{tend} = U_{adv} + U_{cor} + U_{pg} + U_{pbl}$$
$$V_{tend} = V_{adv} + V_{cor} + V_{pg} + V_{pbl}$$ 
(2)

where $U_{tend}$ and $V_{tend}$ are the tendencies of the wind components, $U_{adv}$ and $V_{adv}$ are the advection wind components, $U_{cor} = V$ and $V_{cor} = -U$ are the Coriolis wind components, $U_{pg} = -V_g$ and $V_{pg} = U_g$ are the pressure gradient wind components and $U_{pbl}$ and $V_{pbl}$ are the turbulent diffusion wind components. In a meso-micro offline coupled model, the RANS equations are

solved using mesoscale forcing as source terms in the microscale model. In horizontally homogeneous conditions:

$$\frac{1}{f_c}\frac{\partial U}{\partial t} = U_{adv} + V + U_{pg} - \frac{1}{f_c}\frac{\partial uw}{\partial z} + U_{nud}$$
$$\frac{1}{f_c}\frac{\partial V}{\partial t} = V_{adv} - U + V_{pg} - \frac{1}{f_c}\frac{\partial vw}{\partial z} + V_{nud}$$ 
(3)

where the advection and pressure gradient wind components are derived from mesoscale simulations and vary with the time $t$ and the height above ground level $z$. Bias-correction nudging terms, $U_{nud}$ and $V_{nud}$, have also been incorporated to assimilate profile observations available from a reference measurement campaign. Observational nudging (or Newtonian relaxation)

based on Stauffer and Seaman (1990) is defined as:

$$\delta_{nud} = \frac{\omega_z}{f_c}\frac{(\delta_{obs} - \delta)}{\tau_{nud}},$$ 
(4)

where $\delta_{nud}$ is either $U_{nud}$ or $V_{nud}$, $\delta_{obs}$ and $\delta$ are the corresponding observed and simulated quantities, $\tau_{nud}$ is the nudging time-scale and $\omega_z$ is a weight function that is equal to 1 within the vertical range of the observations, $z_1 < z < z_2$, decreases linearly from 1 to 0 in the range $z_2 < z < 2z_2$ and 0 elsewhere. Since the nudging term is an artificial forcing, it should not be

dominant compared to the other terms in Eq. (3). Hence, it should be scaled by a the time constant $\tau_{nud}$ of the order of the




slowest physical process of the ABL which, for a diurnal cycle, is the inertial oscillation introduced by the Coriolis term. Hence $\tau_{nud}$ should be of the order of $1/f_c$. In general $\tau_{nud}$ is typically between $10^3$ and $10^4$ s in meteorological systems (Stauffer and Seaman, 1990).

Similarly to the momentum equations, the energy equation in the absence of radiative and phase-change heat transfer effects relates the tendency of potential temperature with the mesoscale advective temperature ($\Theta_{adv}$), the diffusion and the nudging ($\Theta_{nud}$) terms.

$$\frac{\partial \Theta}{\partial t} = \Theta_{adv} - \frac{\partial w\theta}{\partial z} + \Theta_{nud} \tag{5}$$

where $w\theta$ is the kinematic heat flux and $\Theta_{nud}$ is defined in Eq. (3).

The diffusion terms in Eqs. (1), (3) and (5) are simulated assuming a isotropic eddy viscosity that relates turbulent fluxes with the gradients of mean flow quantities:

$$uw = K_m \frac{\partial U}{\partial z}; \quad vw = K_m \frac{\partial V}{\partial z}; \quad w\theta = \frac{K_m}{\sigma_t} \frac{\partial V}{\partial z}, \tag{6}$$

where the Prandtl number $\sigma_t$ is assumed to be equal to 1. The eddy viscosity $K_m$ is equivalent to the product of a mixing length and velocity scales. Three turbulent closures will be used in this paper: 1st order, based on an analytical function of the mixing length and a velocity scale based on the strain rate (S-l) (Sanz Rodrigo and Anderson, 2013); 1.5th order, based on the same mixing length function and a velocity scale based on a transport equation of the turbulent kinetic energy (k-l) (Sanz Rodrigo and Anderson, 2013); and 2nd order, based on two transport equations for the turbulent dissipation rate and the turbulent kinetic energy (k-ε) (Sogachev et al., 2012; Koblitz et al., 2013).

The S-l turbulence model assumes a semi-empirical analytical expression for the turbulent mixing length $l_m$:

$$l_m = \frac{\kappa z}{\phi_m(\zeta) + \frac{\kappa z}{\lambda}}, \tag{7}$$

and scales the mixing velocity with the strain rate to obtain the eddy viscosity:

$$K_m = l_m^2 \left[ \left(\frac{\partial U}{\partial z}\right)^2 + \left(\frac{\partial V}{\partial z}\right)^2 \right]^{1/2}, \tag{8}$$





where $\kappa = 0.41$ is the von Karman constant, $\lambda = 0.00037\,S_{g0}/|fc|$ is the maximum mixing length in neutral conditions, proportional to the surface pressure gradient (Blackadar, 1962). $\phi_m$ is an empirical function that depends on the local stability parameter $\zeta = z/L$ based on the Obukhov length $L$. Functional relationships from Dyer (1974) are commonly used:

$$\phi_m(\zeta) = \begin{cases} (1-5\zeta)^{-1/4} & \zeta < 0 \\ 1+5\zeta & \zeta \geq 0 \end{cases}.$$
(9)

5 Transport equations for the turbulent kinetic energy $k$ and dissipation rate $\varepsilon$ are:

$$\frac{\partial k}{\partial t} = P + B - \varepsilon + \frac{\partial}{\partial z}\left(\frac{K_m}{\sigma_k}\frac{\partial k}{\partial z}\right),$$
(10)

$$\frac{\partial \varepsilon}{\partial t} = \frac{\varepsilon}{k}\left(C_{\varepsilon 1}^{*}P - C_{\varepsilon 2}\varepsilon + C_{\varepsilon 3}B\right) + \frac{\partial}{\partial z}\left(\frac{K_m}{\sigma_\varepsilon}\frac{\partial \varepsilon}{\partial z}\right),$$
(11)

where $\sigma_k$ and $\sigma_\varepsilon$ are the Schmidt numbers for $k$ and $\varepsilon$, $P$ and $B$ are the rate of shear and buoyancy production of $k$, and $C_{\varepsilon 2}$ and $C_{\varepsilon 3}$ are model coefficients.

10 Then, the eddy viscosity is defined as:

$$K_m = l_m \frac{k^{1/2}}{C_\mu^{1/4}},$$
(12)

for the $k$-$l$ model and

$$K_m = C_\mu^{1/4} l_m k^{1/2} = C_\mu \frac{k^2}{\varepsilon},$$
(13)

for the $k$-$\varepsilon$ model, where $C_\mu$ is a coefficient equal to the square of the ratio of the shear stress and $k$ in equilibrium.

15 Sogachev et al. (2012) define a modified $C_{\varepsilon 1}$ coefficient as follows:

$$C_{\varepsilon 1}^{*} = C_{\varepsilon 1} + (C_{\varepsilon 2} - C_{\varepsilon 1})\frac{l_m}{l_{max}},$$
(14)

with a length-scale limiter following Mellor and Yamada (1974):

$$l_{max} = C_\lambda \frac{\int_0^\infty z k^{1/2}\,dz}{\int_0^\infty k^{1/2}\,dz},$$
(15)





where $C_\lambda = 0.075$ in order to obtain Blackadar's $l_{max} = \lambda$ in neutral conditions, consistent with Apsley and Castro (1997). Sogachev et al. (2012) introduce a rather complex additional diffusion term in the Eq. (11) to make the $k$-$\varepsilon$ model equivalent to a $k$-$\omega$ model. For simplicity, this term is not included here.

In neutral conditions, a relationship amongst $k$-$\varepsilon$ coefficients is prescribed in order to obtain consistency with Monin-Obukhov theory in the surface layer (Richards and Hoxey, 1993):

$$\sigma_\varepsilon = \frac{\kappa^2}{C_\mu^{1/2}(C_{\varepsilon 2} - C_{\varepsilon 1})} .$$ (16)

In non-neutral conditions, Sogachev et al. (2012) introduce a $C_{\varepsilon 3}$ coefficient that depends on the local stability conditions:

$$C_{\varepsilon 3} = (C_{\varepsilon 1} - C_{\varepsilon 2})\alpha_B + 1,$$ (17)

with:

$$\alpha_B = \begin{cases} 1 - l_m/l_{max} & \text{if } Ri > 0 \\ 1 - \left[1 + (C_{\varepsilon 2} - 1)/(C_{\varepsilon 2} - C_{\varepsilon 1})\right] l_m/l_{max} & \text{if } Ri < 0 \end{cases},$$ (18)

where $Ri = B/P$ is the local gradient Richardson number. With the relationships of Eqs. (16) and (17), the following set of model coefficients are used: $C_{\varepsilon 1} = 1.52$, $C_{\varepsilon 2} = 1.833$, $\sigma_k = 2.95$, $\sigma_\varepsilon = 2.95$ and $C_\mu = 0.03$.

Surface boundary conditions are defined based on MOST using the simulated surface-layer friction velocity $u_{*0}$ and heat flux $w\theta_0$. The potential temperature at the surface $\Theta_0$ is either prescribed or inferred from the 2-m temperature $\Theta_2$:

$$\Theta_0 = \Theta_2 - \frac{\theta_{*0}}{\kappa}\left[\lg\left(\frac{2}{z_{0t}}\right) + \Psi_h\left(\frac{2}{L_0}\right)\right]; \quad \text{with } \theta_{*0} = -\frac{w\theta_0}{u_{*0}},$$ (19)

where a thermal roughness length $z_{0t} = z_0/100$ (Bosveld et al., 2014a) and Dyer's integral form of the stability function for heat $\psi_h(\zeta)$ are adopted.

## 3 Verification

### 3.2 GABLS1: Idealized quasi-steady stable ABL

The GABLS1 case set-up is described in Cuxart et al. (2006), based on LES simulations presented by Kosovic and Curry (2000), where the boundary-layer is driven by a prescribed uniform geostrophic wind and surface cooling rate over a horizontally homogeneous ice surface. The following initial and boundary conditions are used: $f_c = 1.39 \times 10\text{-}4 \text{ s}^{-1}$; $U_g = 8 \text{ m s}^{-1}$; $V_g = 0$; $\Theta_0 = 265 \text{ K}$ for the first 100 m and then increasing at $\Gamma = 0.01 \text{ K m}^{-1}$; $k = 0.4(1 - z/250)^3 \text{ m}^2 \text{ s}^{-2}$ for the first 250 m



and a minimum value of $10^{-9}$ m2 s$^{-2}$ above. The surface temperature $\Theta_0$ starts at 265 K and decreases at a cooling rate of 0.25 K h$^{-1}$. The roughness length for momentum and heat is set to $z_0 = 0.1$ m.

Single-column model simulations are run for 9 hours using a 1-km long log-linear grid of 301 points and a time-step of 1 s (Sanz Rodrigo and Anderson, 2013). Fig. 1 (left) shows surface fluxes and boundary-layer height, based on shear stress, for the three turbulence models and compared with the *k-l* model of Weng and Taylor (2006) and LES simulations from Beare et al. (2006). Fig. 2 shows the quasi-steady profiles resulting at the end of the 9-hr cooling. The three models are consistent with the reference simulations. While the *S-l* and *k-l* models produce almost identical results, the *k-ε* model produces slightly smaller surface momentum flux leading to a slightly lower boundary-layer height. Nevertheless, the differences are small.

A sensitivity analysis of quasi-steady ABL profiles is shown in Fig. 3, following the same simulation approach than GABLS1 and varying the surface cooling rate *CR* and the geostrophic wind $S_g$. In order to use a more representative wind energy context, the inputs correspond to the Fino-1offshore site conditions, with: $f_c = 1.2 \times 10\text{-}4$ s$^{-1}$ and $\Gamma = 0.001$ K m$^{-1}$. The roughness length is proportional to the square of the surface friction velocity through the Charnock relation (Sanz Rodrigo, 2011), with $z_0 = 0.0002$ m being a representative value. Contours of quantities of interest are presented at a reference 'hub-height' of 70 m and a reference 'rotor range' between 33 and 90 m. The stability parameter *z/L* at the reference height is also plotted following the stability classes defined in Sanz Rodrigo et al. (2014). In unstable conditions the boundary-layer height is of the order of 1 km and the reference wind speed is almost independent of the cooling rate. Turbulence decreases and wind shear increases as neutral conditions are approached. In stable conditions the boundary layer height is of the order of a few hundred meters and the wind conditions are more strongly correlated to the local stability parameter. In very stable conditions turbulence is low and a LLJ develops with high shear.

It is important to note that the quasi-steady profiles resulting from the sensitivity analysis are almost never happening in real conditions. They are canonical cases that help us parameterize the ABL without dynamical effects so that we can more easily study the relationship between the main drivers of the ABL. In real conditions, the ABL is a transient phenomena that not only depends on the actual boundary conditions but also on the hours to days of history leading to them.

### 3.2 GABLS2: Idealized diurnal cycle

While the second GABLS exercise was more strongly based on observations, from the CASES-99 experiment in Kansas, from the ABL forcing perspective it can still be regarded as idealized. The case corresponds to two consecutive clear and dry days with a strong diurnal cycle. Since the focus of the study was the intercomparison of boundary-layer schemes, the forcing conditions were simplified to facilitate the comparison among the various turbulent closures, rather than an assessment of their accuracy against the actual observations.



The case set-up and model intercomparison is described in Svensson et al. (2011). The boundary conditions are prescribed in terms of a uniform geostrophic wind of $S_g = 9.5$ m s-1 and a prescribed surface temperature derived from observations. The roughness lengths are set to $z_0 = 0.03$ m and $z_{0t} = z_0/10$. A small subsidence linearly increasing with height up to -0.005 m s$^{-1}$ at 1000 m is also introduced but it will be neglected here for simplicity. For the same reason, humidity will not be modelled

here since its effect on wind profiles is not significant. Initial profiles are defined at 16:00 local time of the 22nd of October 1999 and the simulation runs for 59 hours. The target evaluation day in the GABLS2 benchmark was the 23rd of October. This leaves only 8 hours of spin-up time before the target day for the models to reach equilibrium with the initial conditions. Koblitz et al. (2013) indicate that this short spin-up period is not enough for the diurnal cycle to reach equilibrium with the boundary conditions. An alternative approach is to run a periodic diurnal cycle for several days until equilibrium is reached,

i.e. two consecutive days show the same diurnal cycle. This cyclic approach is also followed here, based on the 48-hr period of surface temperature shown in Fig. 4. After 5 cycles, the maximum difference in potential temperature with the forth cycle is 0.2 K and the velocity field is in equilibrium. A 4-km log-linear grid of 301 points is used with a time step of 1 s.

Fig. 1 (right) shows the surface fluxes and stability parameter of the three turbulence models compared with the SCM results of the GABLS2 model intercomparison of Svensson et al. (2011) and the LES results of Kumar et al. (2010). The three

models are within the scatter of the SCM reference results and close to the LES results. Comparing with the LES simulations, the $k$-$\varepsilon$ model overpredicts the heat flux in unstable conditions and in stable conditions over the second night. Fig. 5 shows time-height contour plots of mean velocity, turbulent kinetic energy and potential temperature for the three models. As the closure order is increased, the models become more dissipative (higher turbulent kinetic energy). Higher mixing during diurnal unstable conditions results in a faster evening transition to nocturnal stable conditions and a higher

LLJ, i.e. lower wind shear in the rotor area.

### 4 Validation

#### 4.1 GABLS3: Real diurnal cycle

The GABLS3 set-up is described in Bosveld et al. (2014a). The case analyzes the period from 12:00 UTC 1 July to 12:00 UTC 2 July 2006, at the Cabauw Experimental Site for Atmospheric Research (CESAR), located in the Netherlands

(51.971ºN, 4.927ºE), with a distance of 50 km to the North Sea at the WNW direction (van Ulden and Wieringa, 1996). The elevation of the site is approximately -0.7 m, surrounded by relatively flat terrain characterized by grassland, fields and some scattered tree lines and villages (Fig. 6). The mesoscale roughness length for the sector of interest (60º - 120º) is 15 cm.

The CESAR measurements are carried out at a 200-m tower, free of obstacles up to a few hundred meters in all directions. The measurements include 10-min averaged vertical profiles of wind speed, wind direction, temperature and humidity at




heights 10, 20, 40, 80, 140 and 200 m, as well as surface radiation and energy budgets. Turbulence fluxes are also monitored at four heights: 3, 60, 100 and 180 m. A RASS profiler measures wind speed, wind direction and virtual temperature above 200 m.

The selection criteria for GABLS3 consisted on the following filters applied to a database of 6 years (2001 - 2006): stationary synoptic conditions, clear skies (net longwave cooling > 30 W m$^{-2}$ at night), no fog, moderate geostrophic winds (5 to 19 m s$^{-1}$, with less than 3 m s$^{-1}$ variation at night) and small thermal advective tendencies. Out of the 9 diurnal cycles resulting from this filtering process, the one that seemed more suitable was finally selected: 12:00 UTC 1 July to 12:00 UTC 2 July 2006.

**4.2 Mesoscale forcing from WRF**

Mesoscale forcing is derived from simulations with the Advanced Research Weather Forecasting model (WRF), version 3.8 (Skamarock et al., 2008). Kleczek et al. (2014) made a sensitivity study of WRF for different grid set-ups, boundary-layer schemes, boundary conditions and spin-up time. Reasonably good results of the vertical wind profile in stable conditions (at midnight) are obtained although the dependency on the PBL scheme and grid set-up is important.

Mesoscale simulations are reproduced here using the same domain set-up used as reference by Kleczek et al., based on three
concentric square domains centred at the Cabauw site. The model is driven by 6-hourly ERA Interim reanalysis data from ECMWF (European Centre for Medium-Range Weather Forecasts), which comes at a resolution of approximately 80 km. Three domains, all with 183x183 grid points, are nested at horizontal resolutions of 9, 3 and 1 km. The vertical grid, approximately 13 km high, is based on 46 terrain-following (eta) levels with 24 levels in the first 1000 m, the first level at approximately 13 m, a uniform spacing of 25 m over the first 300 m and then stretched to a uniform resolution of 600 m in
the upper part. The U.S. Geological Survey (USGS) land-use surface model, that comes by default with the WRF model, is used together with the unified Noah land-surface model to define the boundary conditions at the surface. Other physical parameterizations used are: the rapid radiative transfer model (RRTM), the Dudhia radiation scheme and the Yonsei University (YSU) first-order PBL scheme. The WRF set-up follows the reference configuration of Kleczek et al. except for the input data (Kleczek et al. uses ECMWF analysis), the horizontal resolution (Kleczek et al. use 27, 9 and 3 km) and the
vertical grid (Kleczek et al. uses 34 levels, 15 in the lowest 1000 m). Differences in the grid settings are due to a further study with additional nested domains with large-eddy simulation to study turbulent processes in the ABL. Following Kletzeck et al., we use a spin-up time of 24 hours, i.e. the model is initialized one day before the target evaluation day in order to allow enough time to develop mesoscale processes in equilibrium with the initial and boundary conditions of the reanalysis data.





To derive mesoscale forcing, the momentum budget components (also called tendencies) are directly extracted from WRF as they are computed by the solver (Lehner, 2012). Curvature and horizontal diffusion tendencies have been neglected since they are comparatively small with respect to the other terms of the momentum budget. Fig. 7 shows contour plots of the longitudinal wind component and the momentum budget terms of Eq. (2). These quantities have been spatial and temporal

averaged to filter out microscale fluctuations. The spatial filter is based on 4x4 grid points surrounding the site from the second WRF domain, which defines a typical size of a microscale domain ($L_{avg}$ = 9 km square box). A centred rolling average of $t_{avg}$ = 60 min is also applied in order to remove high frequency fluctuations in the lower part of the boundary layer.

Fig. 8 shows the effect of $L_{avg}$ on the mesoscale forcing, vertically averaged over a 40-200 m layer, which is approximately

the span of a large wind turbine of 8 MW (diameter $D$ = 160 m, hub height $z_{hub}$ = 120 m). If site interpolated values are used ($L_{av}$ = 0 km), large fluctuations can be observed in the mesoscale forcing during convective conditions at the beginning of the cycle. Here, the fluctuations are filtered out when a spatial averaging of $L_{avg}$ = 9 km is introduced, which indicates that the scale of these disturbances are smaller than this size. Extending the spatial averaging to $L_{avg}$ = 30 km does not show significant variations with respect to the 9 km case. It is interesting to note that even though the mean wind speed profiles

does not show any dependency on the spatial averaging, and one could conclude that horizontally homogeneous conditions prevail, there is a quite significant spatial variability of mesoscale forcing within the averaging box.

The derived mesoscale forcing is consistent with that obtained by Bosveld et al. (2014a), based on simulations with the RACMO model at a horizontal resolution of 18 km. In order to facilitate the implementation and interpretation of the mesoscale forcing in the GABLS3 SCMs intercomparison, simplified mesoscale forcing was defined by adjusting piecewise

linear approximations of the RACMO tendencies to obtain a reasonable agreement of the wind speed at 200m.

Even though the filtering process, the resulting smooth fields in Fig. 7 still show large mesoscale disturbances in the advective tendencies, especially during nighttime conditions at greater heights where vertical diffusion is low. The geostrophic wind is more uniform, showing some decrease of intensity with height (baroclinicity). At rotor level (Fig. 8) the pressure gradient force is quite stationary throughout the whole cycle with a sudden change of 50° in wind direction

happening a midnight. The advective wind speed peaks at this time reaching similar values than the geostrophic wind. Interestingly, the advective wind direction makes a 360° turn throughout the cycle.

The dynamical origin of the nocturnal low-level jet was originally described by Blackadar (1957) as an inertial oscillation that develops in flat terrain due to rapid stabilization of the ABL during the evening transition under relatively dry and cloud-free conditions. The daytime equilibrium of pressure gradient, Coriolis and frictional forces is followed by a sudden decrease

of vertical mixing due to radiative cooling during the evening transition. The residual mixed layer in the upper part of the ABL is decoupled from the surface and the Coriolis force induces an oscillation in the wind vector around the geostrophic





wind, producing an acceleration of the upper air which is manifested as a low-level jet at relatively low heights. At Cabauw this happens 20% of the nights with jet heights between 140 and 260 m and jet speeds of 6-10 m s$^{-1}$ (Baas et al., 2009).

**4.3 Quantities of interest**

Revisiting the GABLS3 in wind energy terms means evaluating the performance of the models with application-specific quantities of interest (QoIs). These quantities are evaluated across a reference rotor span of 160 m, between 40 and 200 m, characteristic of a 8 MW large wind turbine. Besides hub-height wind speed $S_{hub}$ and direction $WD_{hub}$, it is relevant to consider the rotor equivalent wind speed *REWS*, the turbulence intensity (not evaluated here), the wind speed shear $\alpha$, and the wind direction shear or veer $\psi$.

The rotor equivalent wind speed is specially suitable to account for wind shear in wind turbine power performance tests (Wagner et al., 2014). The *REWS* is the wind speed corresponding to the kinetic energy flux through the swept rotor area, when accounting for the vertical shear:

$$REWS = \left[ \frac{1}{A} \sum_i \left( A_i S_i^3 \cos \beta_i \right) \right]^{1/3},$$ (20)

where $A$ is the rotor area and $A_i$ are the horizontal segments that separate vertical measurement points of horizontal wind speed $S_i$ across the rotor plane. The *REWS* is weighted here by the cosine of the angle $\beta_i$ of the wind direction $WD_i$ with respect to the hub-height wind direction to account for the effect of wind veer.

Wind shear is defined by fitting a power-law curve across the rotor wind speed points $S_i$:

$$S_i = S_{hub} \left( \frac{z_i}{z_{hub}} \right)^{\alpha}.$$ (21)

Similarly, wind veer is defined as the slope $\psi$ of the linear fit of the wind direction difference with respect to hub-height:

$$\beta_i = \psi \left( WD_i - WD_{hub} \right).$$ (22)

In order to evaluate simulations and measurements consistently, these quantities are obtained after resampling, by linear interpolation, velocity and wind direction vertical profiles at 10 points across the rotor area and then computing the *REWS* and the shear functional fits. While these fitting functions are commonly used in wind energy, their suitability in LLJ conditions is questionable. The regression coefficient from the fitting can be used to determine this suitability.





### 4.4 Metrics

Validation results can be quantified based on the mean absolute error *MAE* metric:

$$MAE = \frac{1}{N}\sum_{i=1}^{N}\left|\chi_{pred} - \chi_{obs}\right|, \tag{23}$$

where $\chi$ is any of the above mentioned QoIs, predicted (pred) or observed (obs), and $N$ is the number of samples evaluated in
the time series.

It is important to note that the errors computed here are particular for this diurnal cycle test case and cannot be associated to
the general accuracy of the SCM in other situations. It is more important to discuss the results in relative terms to explain, for
instance, the impact of adding modeling complexity as we go from idealized to more realistic forcing. Then, if a simulation
is used as a reference to quantify this relative improvement, it is convenient to use a normalized *MAE* by dividing with
respect to the *MAE* of the reference simulation:

$$NMAE = \frac{MAE}{MAE_{ref}}. \tag{24}$$

### 4.5 Results

Table 1 shows a list of the simulations performed with the single-column model using different settings in terms of surface
boundary conditions and mesoscale forcing. The SCM simulations have been run with the same grid set-up of GABLS2, i.e.
4-km long log-linear grid with 301 levels and a time step of 1 s. The simulations are grouped according to different model
evaluation objectives as described in the last column of Table 1. Table 2 shows the *MAE* and normalized *MAE,* with respect
to the reference *k-ε* SCM simulation (*ke_T2*: tendencies from WRF, no nudging, surface boundary conditions based on
prescribed WRF 2-m temperature), for the rotor-based QoIs integrated throughout the diurnal cycle. Time-series of these
QoIs are plotted in Fig. 12 and surface fluxes in Fig. 11. ERA-Interim and WRF simulations are included in the plots in
order to show how the mesoscale model transforms the inputs from the reanalysis data and then is used as input to the
microscale model simulations in the meso-micro model-chain.  As we did with the mesoscale forcing, a centred rolling
average of 60 min is applied to simulations and observations in order to have all the quantities evaluated on a common
timeframe.

#### 4.5.1 Consistency of mesoscale tendencies and nudging bias-correction methods from a model-chain perspective

Fig. 9 shows time-height contour plots of wind velocity, wind direction and potential temperature for the WRF simulation,
reference SCM simulation without nudging (*ke_T2*) and with wind speed nudging between 40 and 200 m (*UVnud200_tau10*)
and the observations. The reference rotor span, between 40 and 200 m, is delimited with dashed lines. By comparing the first





two columns of Fig. 9 we can see that the SCM shows similar footprint as the mesoscale model even though the simplified physics used. In terms of *REWS*, the MAE due to offline coupling is only 4% of the error of the WRF model itself. (Table 2). This confirms the consistency of the asynchronous coupling methodology based on mesoscale tendencies. Comparing with observations, we can distinguish a LLJ of longer duration in the simulations than in the models, the simulations

showing a double peak while observations show a more distinct velocity maxima. The evening and morning transitions are more gradual in the mesoscale model than in the observations.

At the rotor area, the peak of the *REWS* is well predicted by both the mesoscale and the *ke_T2* SCM while they both tend to overpredict in the convective and transitional parts of the cycle (Fig. 12). The LLJ lives longer in the simulations than in the observations. This is attributed to an incorrect timing of the advection tendencies. Switching off these tendencies in the SCM

sifts the LLJ peak of wind speed and direction 3 hours ahead. Wind shear is not predicted well by the models. The reanalysis data predicts surprisingly well the wind shear but, due to the very coarse vertical resolution of the data, this is consider an artefact from the linear interpolation. Wind veer suffers the consequences of the phase error in the wind direction, underpredicting the maximum wind veer. Wind direction is reasonably well predicted by the reanalysis input data, with a ramp starting at 18:00 UTC 1 July and peaking at 6:00 UTC 2 July. However, the mesoscale model presents a sudden change

around midnight, which is apparent in both the pressure gradient and advective forcing in Fig. 8, and results in a broader wind direction peak. This peak has larger amplitude and shorter duration in the observations. The potential temperature fields are also reasonably well characterized by the input data during daytime conditions. At night the cooling is underpredicted by the reanalysis data but overpredicted by the mesoscale model (Fig. 11).

By introducing profile nudging, these deviations are corrected to a large extent in the lower part of the ABL. Since the

weighting function of the nudging terms $\omega_z$ decays linearly up to 400 m we can see how the bias correction is gradually introduced and the simulation is not affected by nudging in the upper levels (Fig. 9). In terms of *NMAE*, using velocity profile nudging leads to error reductions of up to 80% in the *REWS* with respect to the reference simulation (no nudging). A more detailed assessment of profile nudging for different measurement strategies is discussed later.

### 4.5.2 Choice of turbulence closure

The $k$-$\varepsilon$ closure is chosen as reference with respect to the other turbulence models because it is expected to be more generally applicable in heterogeneous terrain conditions, where the mixing length is modelled through the $\varepsilon$ equation. In the GABLS2 case we could see some differences between the three models in the prediction of turbulent kinetic energy when simulating the CASES-99 diurnal case. Here, we quantify the impact on the choice of turbulence model on the QoIs by using the same boundary conditions and mesoscale forcing. The *S-l* and *k-l* models are almost equivalent but show around 30% higher MAE

than the *k-ε* model. Some improvement, of the order of 10%, is observed for lower-order models in the hub-height wind



direction and wind veer but this does not compensate the error increase of 20% in hub-height wind speed and 40% in wind shear.

### 4.5.3 Choice of surface boundary conditions

The third objective in the model evaluation strategy of Table 1 is to determine if there is a choice of boundary condition for the energy equation that is more adequate in the prediction of QoIs. Basu et al. (2008) demonstrated using MOST arguments that using a prescribed surface heat flux as boundary condition in stable conditions should be avoided. MOST is imposed at the surface by prescribing the mesoscale 2-m temperature ($ke\_T2$), the 2-m temperature and surface heat flux ($ke\_T2wt$) or the surface skin temperature ($ke\text{-}Tsk$). Fig. 11 shows time series of surface-layer fluxes (at 3-m height) and 2-m temperature along the diurnal cycle. It is observed a large bias in the 2-m temperature of the WRF simulation which was also found in the GABLS3 model intercomparison (Bosveld et al., 2014b) and WRF sensitivity study of Kleczek et al. (2014). Using the WRF skin temperature instead of the 2-m temperature is equivalent in terms of predicting the surface-layer fluxes. This is not a surprise since the Noah land surface model in WRF is also based on MOST surface-layer parameterization and the roughness lengths in WRF and SCM simulations are the same. However, in terms of *REWS*, using skin temperature instead of 2-m temperature results in 15% increase of the MAE. Adding the WRF heat flux as an additional prescribed quantity also has no effect on the surface fluxes and little impact on the QoIs.

Interestingly enough, prescribing the observed 2-m temperature instead of the mesoscale 2-m temperature results in 23% increase of *REWS* MAE. This is due to a mismatch between the surface (observed) and top (simulated) boundary conditions that lead to a less accurate prediction of potential temperature gradients throughout the ABL. In effect, even though the large bias in the prediction of the potential temperature, the mesoscale simulation is still doing a good job at simulating the diurnal evolution of vertical potential temperature gradients, which are ultimately the main feedback in the simulation of the wind speed fields via the buoyancy term in the turbulence equations. Then, using the mesoscale 2-m temperature as indirect surface boundary condition seems to be the most appropriate choice. This is a standard output in meteorological models and surface stations so it makes sense to use it for practical reasons also as standard in wind energy campaigns and flow models.

### 4.5.4 Added value of more realistic forcing

Adding mesoscale tendencies to microscale ABL simulations requires the generation of tendencies from a mesoscale model. The question is how important are these tendencies in the assessment of QoIs. This is the fourth objective in the model evaluation strategy of Table 1. The modulation of the LLJ evolution by the mesoscale tendencies in the GABLS3 episode is discussed by Bass et al. (2010) and Bosveld et al. (2014a). They use a SCM to switch on and off different forcing mechanisms and show their relative impact in the evolution of the LLJ. Fig. 10 shows time-height plots of different SCM simulations: with all mesoscale tendencies included ($T2\_ke$), without $\Theta_{adv}$ ($noTadv$), without $\Theta_{adv}$, $U_{adv}$ and $V_{adv}$ ($noTadvUadv$) and without advection tendencies and assuming that the geostrophic wind only varies with time following the





surface pressure gradient (*noTadvUadv_Sg0*). The next step in terms of simplifying the forcing would be to impose a uniform geostrophic wind throughout the entire episode, which is the idealized set-up of GABLS2.

In the first 100 m above the ground, where turbulence diffusion is important, advection tendencies are relatively small and using surface geostrophic forcing provides a realistic evolution of the diurnal cycle. Above 100 m advective tendencies

become a dominant force in the modulation of the equilibrium between Coriolis and pressure gradient forces. If only surface geostrophic forcing is applied at greater heights, the wind speed and direction are way off. In terms of the *REWS NMAE*, removing potential temperature tendencies doesn't have a significant impact while additionally removing momentum tendencies results in 24% increase of error. Using just the surface geostrophic wind as forcing increases the error by an additional 100%. Hence, realistic forcing requires the characterization of the horizontal pressure gradient variations with

time and height as main driver. Then, even though advection tendencies come with high uncertainty, introducing mesoscale momentum advection still results in significant improvement. Potential temperature advection in this case shows some improvement in the wind direction and wind shear but this is compensated with a deterioration on wind speed and wind veer, so the overall impact on *REWS* is not significant.

### 4.5.4 Assessment of bias-correction for different profile nudging strategies

In homogeneous terrain conditions, such as those of the GABLS3 case, we should not expect improvements when using the offline meso-micro simulations with a RANS model with respect to online mesoscale simulations with a boundary-layer scheme, since the surface conditions haven't changed and the turbulence models are similar. Instead, by using the same surface conditions, we demonstrated that using mesoscale tendencies was an effective solution to drive a microscale ABL model offline without introducing significant additional uncertainties due to the coupling between the models. It is also not

surprising to find large errors in the WRF model hour-to-hour, sometimes even larger than in the reanalysis input data, since the higher resolution of the model will bring additional variability that is physically realistic but is not necessarily well represented by the models (Bass et al., 2010; Bosveld et al., 2014). In aggregated terms, it has been demonstrated that adding mesoscale-generated advection tendencies was beneficial for the SCM simulations, even though their contribution hour-by-hour was not obvious due to phase errors for instance. A solution to improve the transient behaviour of the microscale model

is to introduce data assimilation through nudging. Here, we explore the profile nudging method of Eq. (4) that depends on the time scale $\tau_{nud}$ and the range and type of observations assimilated in the simulations.

Two scenarios of data assimilation are considered in Table 1 making use of the Cabauw instrumentation as a proxy for typical set-ups that could be used in the wind energy context. The first scenario corresponds to mast-based instrumentation where we can routinely measure and assimilate in the model wind speed and temperature. By convention, temperature

measurements start at 2 m and wind speed measurements at 10 m. Then, the mast height is varied from 80 m (*ke_T2obs_UVTnud80*) to 200 m. Since temperature nudging starts at 2 m, the observed 2-m temperature is prescribed in the





surface boundary condition. By default, the nudging time scale is set to 1 hour. In terms of *REWS*, using data assimilation with an 80 m mast does not improve the aggregated error for a large rotor in the range 40-200 m. Using 120 m or 200 m results in improvements of 12% and 50% respectively. If the time scale is reduced to 10 min, a much stronger correction is introduced every time step and the *REWS* error decreases to almost 90%.

The second scenario corresponds to a lidar set-up whose range typically starts from 40 m and goes up to 200-400 m. Here, only wind speed profiles are assimilated. Again, considering a default nudging time scale of 1 hour, it is observed an improvement of 53% and 58% when assimilating data up to 200 and 400 m respectively. Measuring above the rotor range in this case has little benefit. Comparing the two scenarios, mast or lidar, for a nudging range up to 200 m, it is observed that the main advantage of assimilating potential temperature profiles is in improving the wind shear and veer predictions. This is

also observed at shorter nudging time-scales, particularly during the morning transition (Fig. 12). Fig. 8 shows the magnitude and direction of the nudging correction, vertically averaged over the rotor range and compared to the other forcing terms. Using a nudging time-scale of 60 min results in corrections of less than 1 m s$^{-1}$, comparatively small with respect to the pressure gradient forcing at around 8 m s$^{-1}$. This correction increases occasionally to up to 2 m s$^{-1}$ for a time-scale of 30 min and up to 4 m s$^{-1}$ for a time-scale of 10 min. The direction of the nudging term shows how the correction is mainly following

the advection forcing which comes with higher uncertainty than the pressure gradient force.

Fig. 13 shows the vertical wind profiles of horizontal wind speed and wind direction at midnight and during the morning transition. At midnight, the WRF model is performing reasonably well at developing the nocturnal LLJ and the nudging corrections are mainly affecting the wind direction profile. In contrast, the morning transition is not well captured by the model and large nudging corrections are needed in both wind speed and direction. In both cases, it is apparent the transition

at 400 m between the corrected and uncorrected parts of the profile. Using a linear decaying weight of the nudging correction above 200 m produces a reasonably smooth transition.

**5 Discussion and Conclusions**

The series of GABLS test cases for the evaluation of ABL models have been used for the design of a single-column model that uses realistic forcing by means of mesoscale tendencies and data assimilation at microscale. The model includes three

different turbulent closures that produce consistent results in the idealized cases GABSL 1 and 2. A sensitivity analysis of quasi-steady simulations following the GABLS 1 approach shows how the wind conditions at rotor heights are correlated mostly with the geostrophic wind in unstable conditions and with the local atmospheric stability in stable conditions. The main difference between the models in the GABLS 2 diurnal case resides in a larger turbulent kinetic energy as the order of the closure model is increased.





The GABLS3 diurnal cycle case has been revisited and evaluated in terms of wind energy specific metrics. Instead of using the adjusted mesoscale tendencies of the original GABLS3 set-up, the mesoscale tendencies computed by WRF are directly used to force the SCM. Momentum budget analysis shows the relative importance of the different forcing terms in the momentum equations. By spatial and temporal averaging, the high-frequency fluctuations due to microscale effects are
filtered out. Using mesoscale tendencies to drive the SCM results in consistent flow fields compared to the WRF simulation, even though the more simplified physics of the ABL.

By sensitivity analysis on the mesoscale tendencies, it is shown that the main driver of the ABL is the time and height dependent horizontal pressure gradient. Advection terms come with high uncertainties and hour-to-hour they can lead to large errors. Nevertheless, their impact in terms of QoI's aggregated errors is positive.

The $k$-$\varepsilon$ model of Sogachev et al. (2012) presents better performance than the lower-order turbulence closure models. Considering surface boundary conditions for the potential temperature equation, prescribing the surface temperature by indirectly introducing the WRF 2-m temperature with MOST is more adequate than using the skin temperature or the observed 2-m temperature.

Instead of adjusting at mesoscale, corrections are introduced at microscale through observational profile nudging, to make
use of the routine measurements collected in wind energy campaigns. Mast-based and lidar-based profiler set-ups are compared to show the added value of measuring at greater heights than the hub-height, main advantage of lidar systems. Sensitivity to the nudging time-scale is large, specially to compensate errors introduced by the mesoscale advection forcing.

The GABLS cases show the complexity of interpreting mesoscale forcing. While the pressure gradient force is dominated by large scales and will be reasonably well captured in the reanalysis data, advection tendencies depend on the physical
parameterizations of the mesoscale model. Bass et al. (2010) presented an alternative case based on the ensemble averaging of nine diurnal cycles that meet the GABLS3 selection criteria. This composite case, like the presented GABLS3 case, is entirely based on forcing from a mesoscale model, and facilitates the assessment of the main features of the diurnal cycle by cancelling out the mesoscale disturbances of the individual days. As a result, the composite case shows great improvement versus considering any single day separately. Hence, the assessment of mesoscale to microscale methodologies is more
appropriate in a climatological than in a deterministic sense. Otherwise, dynamical corrections like profile nudging will be required.

**6 Acknowledgements**

This article was produced with funding from the 'MesoWake' Marie Curie International Outgoing Fellowship (FP7-PEOPLE-2013-IOF, European Commission's grant agreement number 624562).



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





**Table 1: List of simulations and objectives for the sensitivity analysis of single-column models.**

| | Turb. | Surface B.C.[1] | Forcing[2] | Objectives |
|---|---|---|---|---|
| *WRF-YSU* | YSU | Noah | ERA Interim | Demonstrate consistency of online (WRF) vs asynchronous meso-micro coupling |
| *ke_T2 (reference)* | $k$-$\varepsilon$ | WRF $T_2$ | WRF tendencies | |
| *Sl_T2* | $S$-$l$ | WRF $T_2$ | WRF tendencies | Evaluate the choice of turbulent closure with realistic forcing |
| *kl_T2* | $k$-$l$ | WRF $T_2$ | WRF tendencies | |
| *ke_T2wt* | $k$-$\varepsilon$ | WRF $T_2$ and $w\theta_0$ | WRF tendencies | Quantify the impact of the choice of surface boundary conditions on fluxes and QoIs |
| *ke_Tsk* | $k$-$\varepsilon$ | WRF $\Theta_0$ | WRF tendencies | |
| *ke_T2obs* | $k$-$\varepsilon$ | Observed $T_2$ | WRF tendencies | |
| *noTadv* | $k$-$\varepsilon$ | WRF $T_2$ | without $\Theta_{adv}$ tendency | Quantify the relative importance of mesoscale tendencies on QoIs |
| *noTadvUadv* | $k$-$\varepsilon$ | WRF $T_2$ | without advection tendencies | |
| *noTadvUadv_Sg0* | $k$-$\varepsilon$ | WRF $T_2$ | only surface pressure gradient | |
| *UVTnud80* | $k$-$\varepsilon$ | Observed $T_2$ | *U,V:* 10-80 m; $\Theta$: 2-80 m; $\tau_{nud}$ = 60 min | Assess bias-correction nudging method using typical wind energy mast configurations |
| *UVTnud120* | $k$-$\varepsilon$ | Observed $T_2$ | *U,V:* 10-120 m; $\Theta$: 2-120 m; $\tau_{nud}$ = 60 min | |
| *UVTnud200* | $k$-$\varepsilon$ | Observed $T_2$ | *U,V:* 10-200 m; $\Theta$: 2-200 m; $\tau_{nud}$ = 60 min | |
| *UVTnud200_tau10* | $k$-$\varepsilon$ | Observed $T_2$ | *U,V:* 10-200 m; $\Theta$: 2-200 m; $\tau_{nud}$ = 10 min | |
| *UVnud400* | $k$-$\varepsilon$ | WRF $T_2$ | *U,V:* 40-400 m, $\tau_{nud}$ = 60 min | Assess bias-correction nudging method using typical wind energy lidar configurations |
| *UVnud200* | $k$-$\varepsilon$ | WRF $T_2$ | *U,V:* 40-200 m, $\tau_{nud}$ = 60 min | |
| *UVnud200_tau30* | $k$-$\varepsilon$ | WRF $T_2$ | *U,V:* 40-200 m, $\tau_{nud}$ = 30 min | |
| *UVnud200_tau10* | $k$-$\varepsilon$ | WRF $T_2$ | *U,V:* 40-200 m, $\tau_{nud}$ = 10 min | |

[1] *All based on Monin-Obkuhov land surface model*

[2] *All use the same WRF tendencies, adding nudging or modified tendencies as indicated*





**Table 2: MAE and normalized MAE with respect to the reference $k\text{-}\varepsilon$ SCM simulation.**

| | REWS [m s⁻¹] | | $S_{hub}$ [m s⁻¹] | | $WD_{hub}$ [º] | | $\alpha$ (shear) | | $\Psi$ (veer) | |
|---|---|---|---|---|---|---|---|---|---|---|
| | MAE | NMAE | MAE | NMAE | MAE | NMAE | MAE | NMAE | MAE | NMAE |
| WRF-YSU | 1.37 | | 1.48 | | 11.59 | | 0.13 | | 0.08 | |
| ke_T2 (reference) | 1.42 | | 1.54 | | 12.72 | | 0.14 | | 0.08 | |
| Sl_T2 | 1.87 | 1.31 | 1.85 | 1.20 | 11.40 | 0.90 | 0.19 | 1.42 | 0.07 | 0.95 |
| kl_T2 | 1.84 | 1.30 | 1.81 | 1.17 | 10.88 | 0.86 | 0.19 | 1.38 | 0.07 | 0.90 |
| ke_T2wt | 1.40 | 0.99 | 1.49 | 0.97 | 12.71 | 1.00 | 0.13 | 0.96 | 0.08 | 1.04 |
| ke_Tsk | 1.63 | 1.15 | 1.91 | 1.24 | 16.39 | 1.29 | 0.15 | 1.10 | 0.10 | 1.29 |
| ke_T2obs | 1.75 | 1.23 | 1.77 | 1.15 | 11.66 | 0.92 | 0.12 | 0.90 | 0.09 | 1.16 |
| noTadv | 1.44 | 1.01 | 1.30 | 0.84 | 13.77 | 1.08 | 0.17 | 1.27 | 0.06 | 0.82 |
| noTadvUadv | 1.76 | 1.24 | 1.87 | 1.22 | 11.78 | 0.93 | 0.18 | 1.31 | 0.07 | 0.96 |
| noTadvUadv_Sg0 | 3.21 | 2.26 | 3.20 | 2.08 | 16.17 | 1.27 | 0.29 | 2.17 | 0.12 | 1.53 |
| ke_T2obs_UVTnud80 | 1.42 | 1.00 | 1.36 | 0.88 | 10.33 | 0.81 | 0.14 | 1.05 | 0.07 | 0.86 |
| ke_T2obs_UVTnud120 | 1.26 | 0.88 | 1.17 | 0.76 | 11.85 | 0.93 | 0.14 | 1.04 | 0.09 | 1.12 |
| ke_T2obs_UVTnud200 | 0.71 | 0.50 | 0.76 | 0.49 | 9.36 | 0.74 | 0.09 | 0.68 | 0.04 | 0.53 |
| ke_T2obs_UVTnud200_tau10 | 0.16 | 0.11 | 0.19 | 0.12 | 3.80 | 0.30 | 0.05 | 0.35 | 0.02 | 0.25 |
| ke_T2_UVnud400 | 0.59 | 0.42 | 0.73 | 0.47 | 10.13 | 0.80 | 0.12 | 0.86 | 0.05 | 0.68 |
| ke_T2_UVnud200 | 0.66 | 0.47 | 0.80 | 0.52 | 10.49 | 0.82 | 0.12 | 0.86 | 0.05 | 0.66 |
| ke_T2_UVnud200_tau30 | 0.45 | 0.31 | 0.49 | 0.32 | 7.21 | 0.57 | 0.10 | 0.73 | 0.05 | 0.59 |
| ke_T2_UVnud200_tau10 | 0.26 | 0.18 | 0.34 | 0.22 | 4.39 | 0.35 | 0.08 | 0.58 | 0.05 | 0.59 |





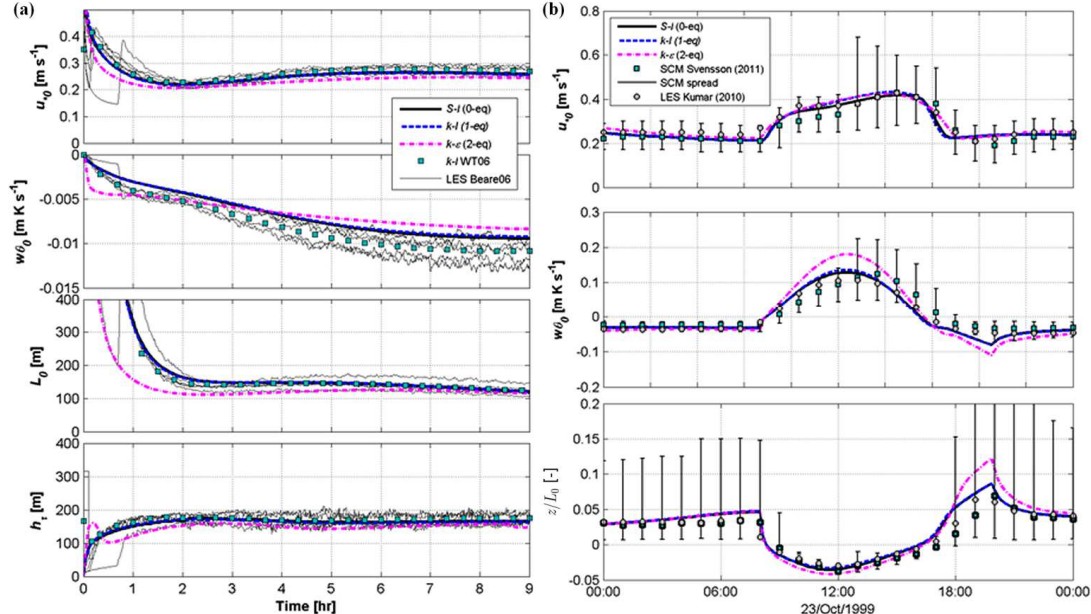

**Figure 1: GABLS1 (a) and GABLS2 (b) time series of boundary-layer height $h_{\tau}$ and surface-layer friction velocity $u_{*0}$, kinematic heat flux $w\Theta_0$, Obukhov length $L_0$ and stability parameter $z/L_0$. Comparison between SCM simulations using three turbulent closures ($S$-$l$, $k$-$l$ and $k$-$\varepsilon$) and the $k$-$l$ model of Weng and Taylor (2006), SCM simulations in Svensson et al. (2011), and LES simulations of Beare et al. (2006) and Kumar et al (2010).**




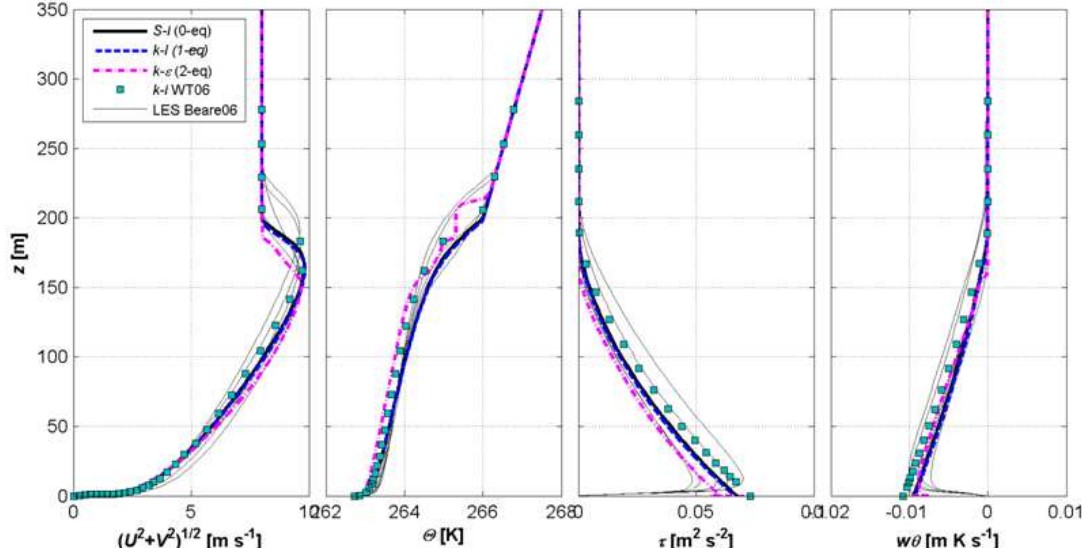

**Figure 2:** GABLS1 quasi-steady vertical profiles of horizontal wind speed $S = (U^2 + V^2)^{1/2}$ potential temperature $\Theta$, shear stress $\tau$ and kinematic heat flux $w\Theta$. Comparison between SCM simulations using three turbulent closures ($S$-$l$, $k$-$l$ and $k$-$\varepsilon$) and the $k$-$l$ model of Weng and Taylor (2006) and the LES simulations of Beare et al. (2006).





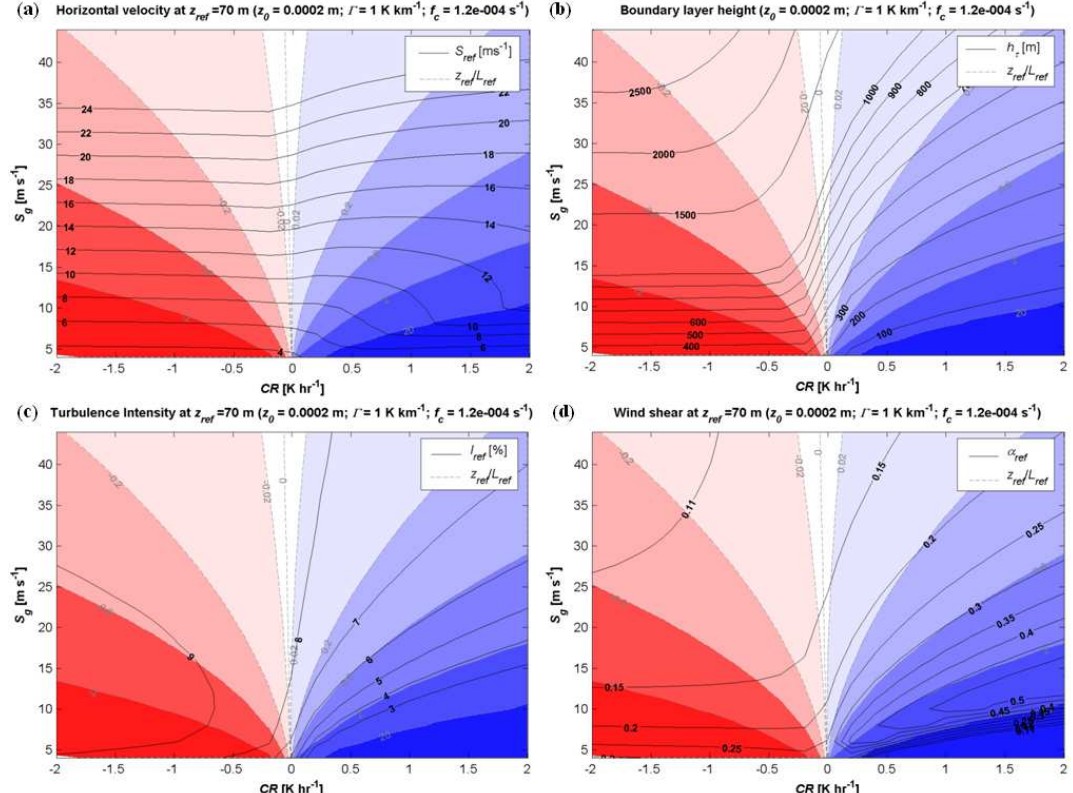

**Figure 3: Sensitivity analysis of quasi-steady profiles at different cooling rates $CR$ and geostrophic wind speed $S_g$ in offshore conditions ($z_0 \sim 0.0002$ m) with an inversion lapse rate of $\Gamma = 1$ K km$^{-1}$. All simulations based on the GABLS1 set-up of 9-hr uniform surface cooling, averaging over the last hour to obtain the quasi-steady profiles. Power-law shear exponent based on 33 and 90 m levels. Atmospheric stability based on the local Obukhov parameter $\zeta = z/L$ at a reference height of 70 m. Stability levels according to Sanz Rodrigo et al. (2014).**



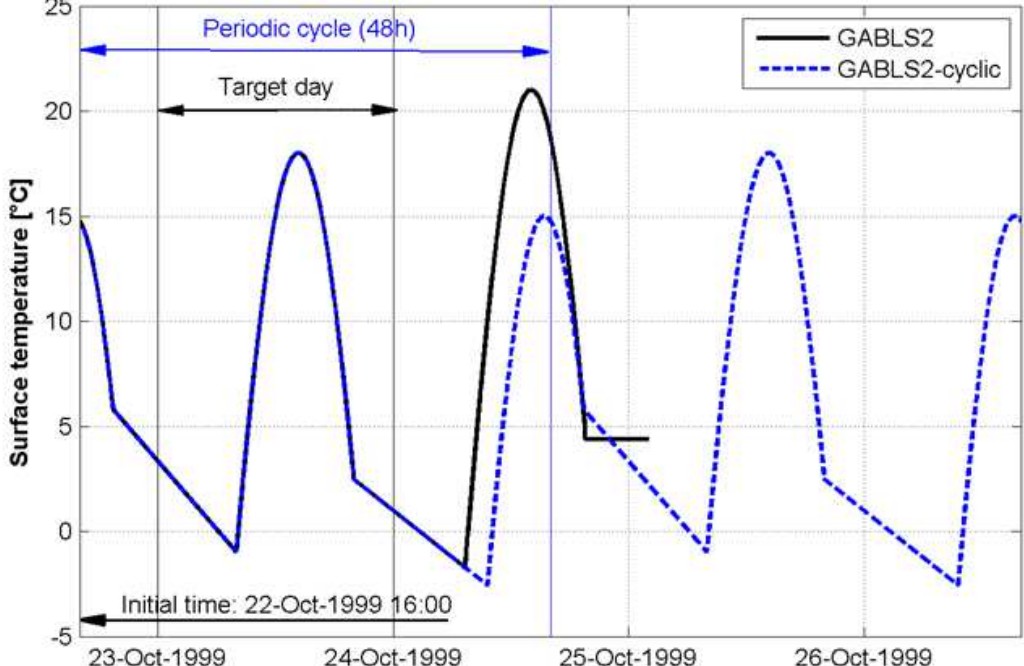

**Figure 4: GABLS2 surface temperature profile (Svensson et al., 2011) and alternative 48-h periodic cycle used to obtain a diurnal cycle independent of initial conditions.**





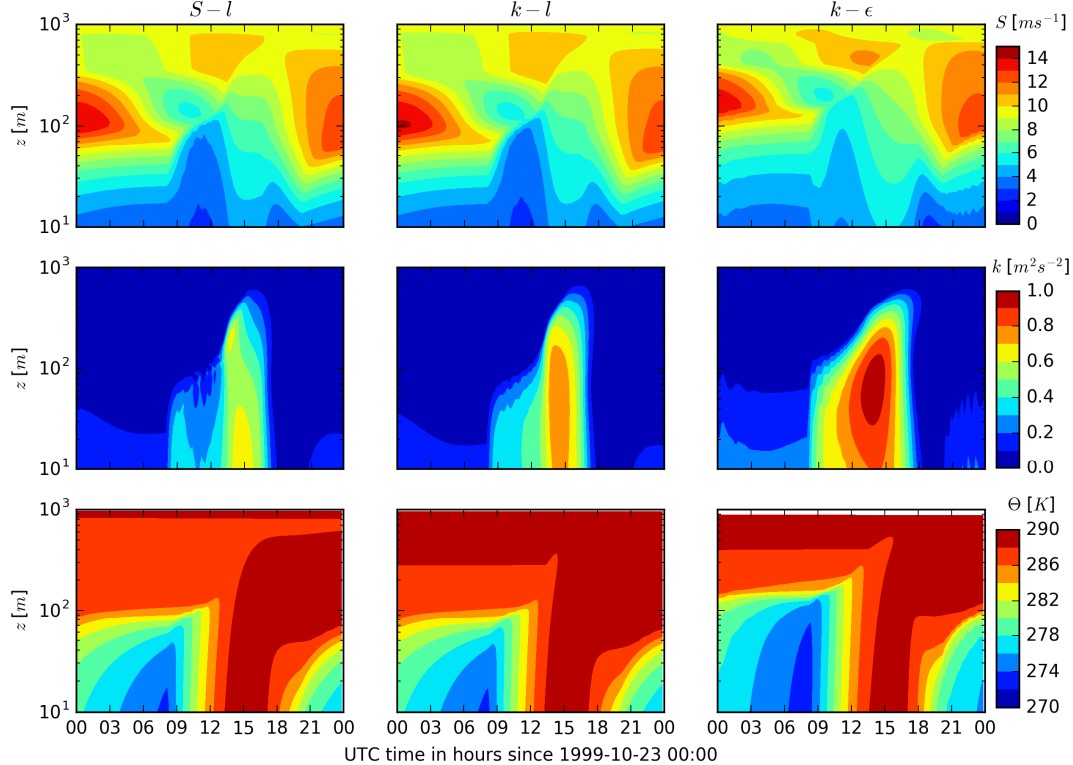

**Figure 5: GABLS2 time-height contour plots of wind velocity _S_ (top raw), turbulent kinetic energy _k_ (middle) and potential temperature _Θ_ (bottom) for the SCM simulation based on _S-l_ (first column), _k-l_ (second) and _k-ε_ (third) turbulence closure after 5 cyclic simulations.**





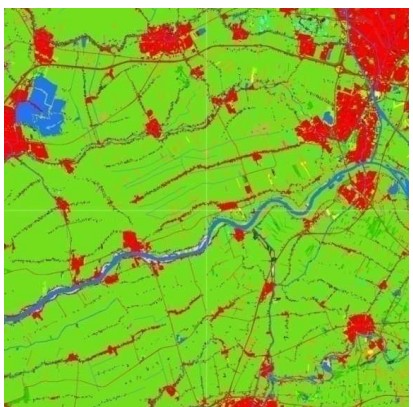

**Figure 6: Roughness map for a 30x30 km area centred at the Cabauw site. Grassland (green) dominates the surface conditions with local values of the roughness length of around 3 cm. For the 60º - 120º sector of interest, the mesoscale roughness length is around 15 cm, characteristic of scattered rough terrain (Verkaik and Holtslag, 2007). This value is also found in the default land-**
5  **use model of WRF, based on the U.S. Geological Survey (USGS, 2011). Figure reprinted from KNMI's Hydra Project website (KNMI, 2016).**





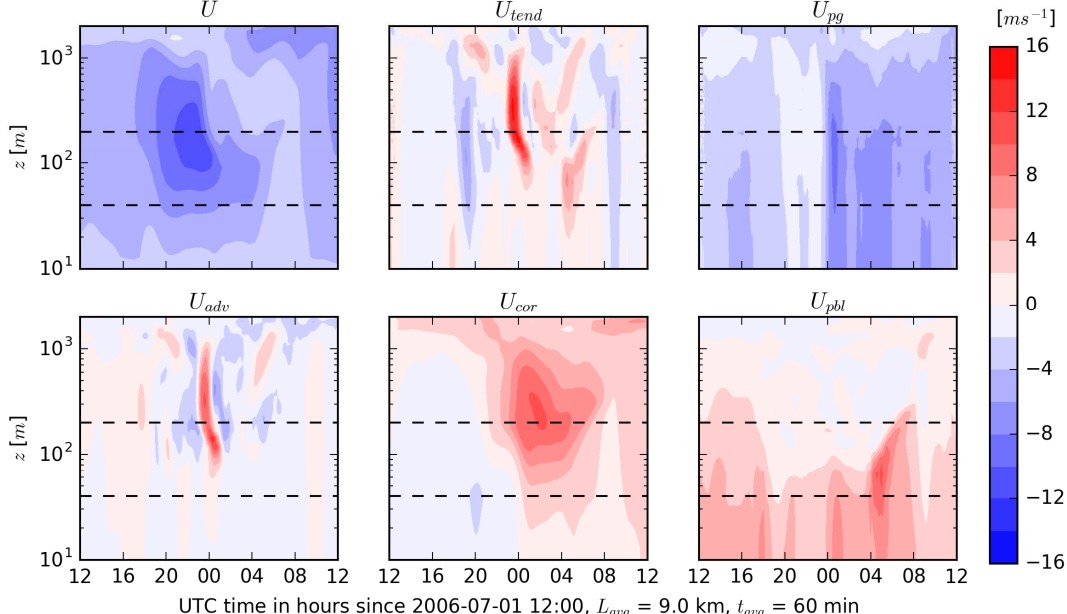

**Figure 7:** Time-height contour plots of the longitudinal wind component $U$ and momentum budget terms: $U_{tend} = U_{adv} + U_{cor} + U_{pg} + U_{pbl}$ from the WRF-YSU simulation.




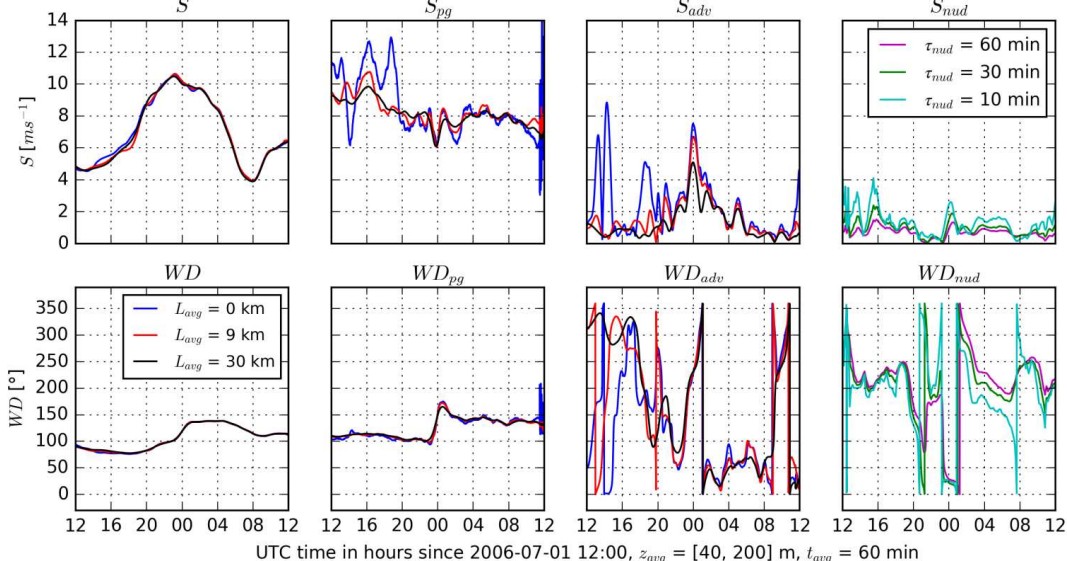

**Figure 8: Magnitude *S* and direction *WD* of the wind vector, pressure gradient, advective and nudging forcing vertically averaged over a rotor span between 40 and 200 m. Sensitivities to spatial averaging *L<sub>avg</sub>* and nudging time-scale *τ<sub>nud</sub>*.**





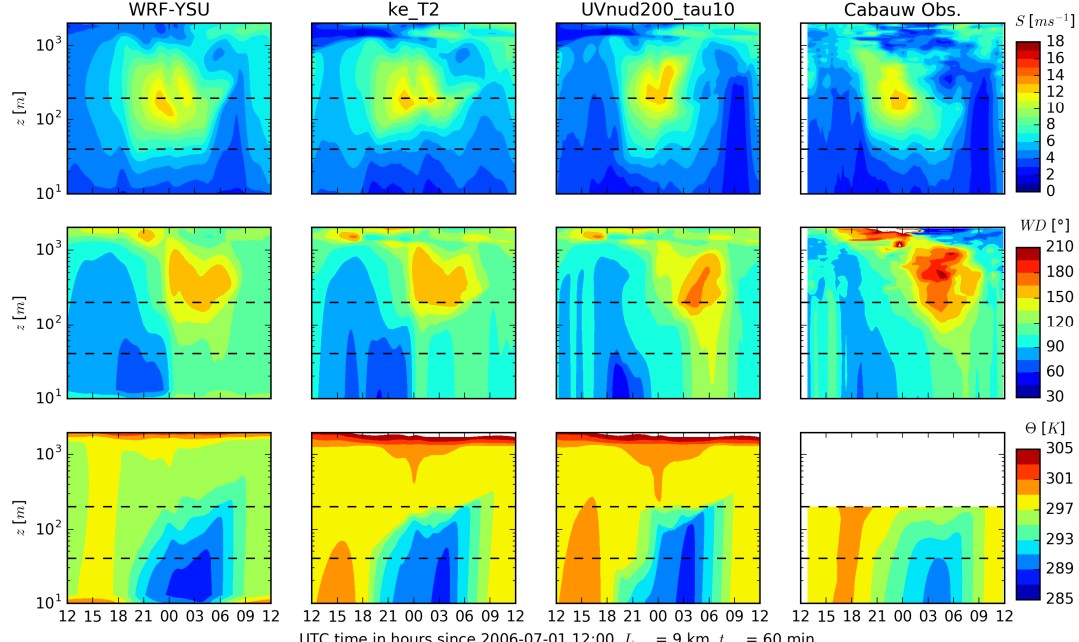

**Figure 9: Time-height contour plots of wind velocity $S$ (top raw), wind direction $WD$ (middle) and potential temperature $\Theta$ (bottom) for the WRF simulation (first column), SCM simulation based on WRF mesoscale forcing and $k$-$\varepsilon$ turbulence closure without (second) and with (third) velocity nudging between 40 and 200 m, and observations (fourth). A reference rotor span (40 - 200 m) is delimited with the dashed lines.**



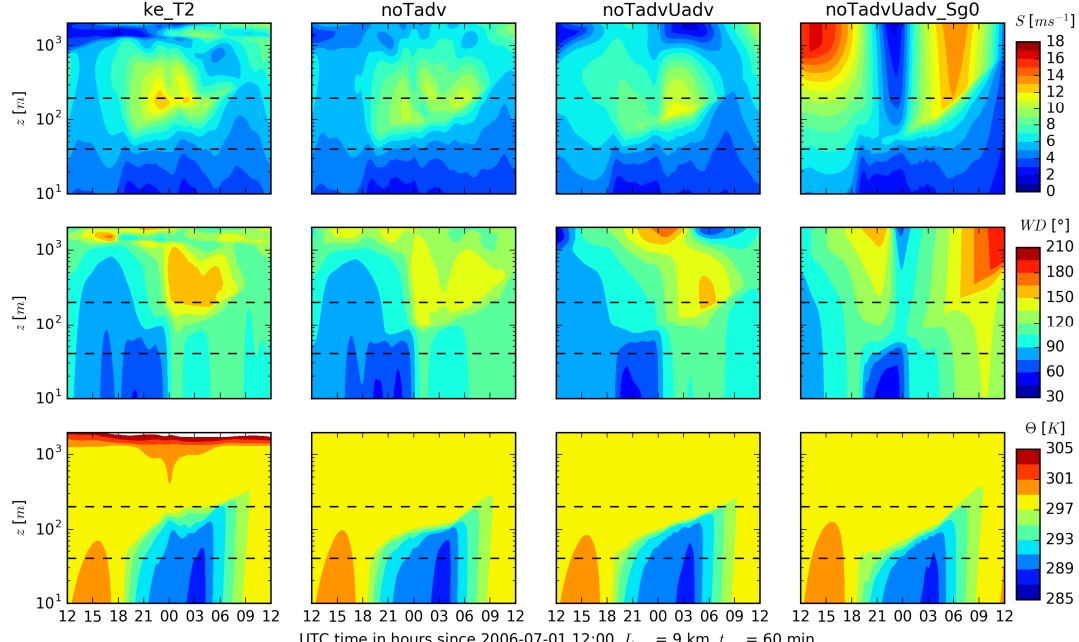

**Figure 10: Time-height contour plots of wind velocity $S$ (top raw), wind direction $WD$ (middle) and potential temperature $\Theta$ (bottom) for four $k$-$\varepsilon$ SCM simulations: with all the forcing terms (first column), without $\Theta_{adv}$(second), without $\Theta_{adv}$ $U_{adv}$ and $V_{adv}$ (third) and without advection and assuming that the geostrophic wind only varies with time following the surface pressure gradient $S_{g0}$ (fourth).**




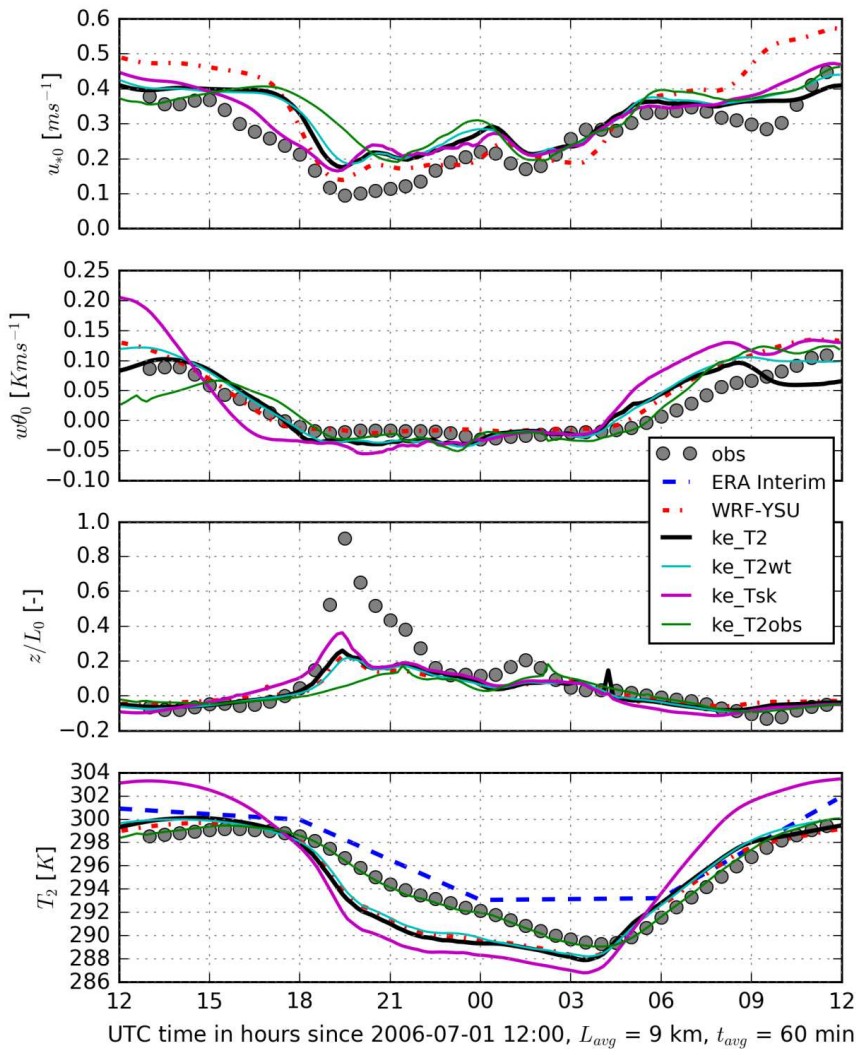

**Figure 11: Time series of surface layer characteristics using different surface boundary conditions for potential temperature with the *k-ε* model and compared with ERA Interim input data, mesoscale model simulation and observations.**





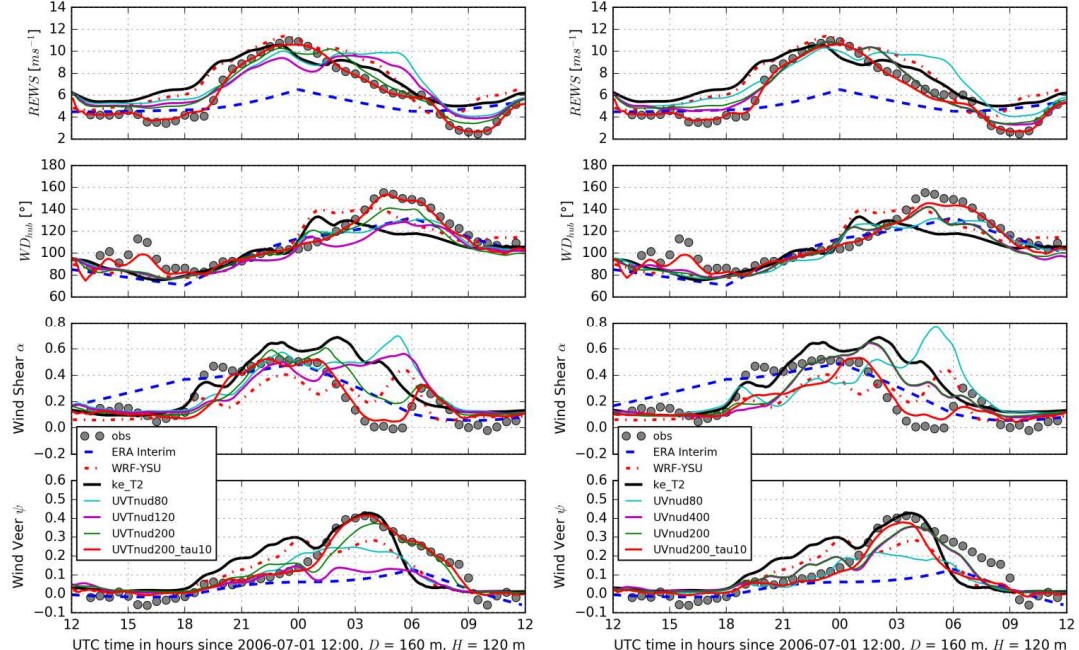

**Figure 12: GABLS3 time series of rotor-based quantities of interest, from top to bottom: rotor equivalent wind speed *REWS*, hub-height wind direction *WD_hub*, wind shear α and wind veer ψ. Sensitivity of the *k-ε* SCM to different nudging strategies, as per Table 1, assimilating wind speed observations "UV" (left) and wind speed and air temperature observations "UVT" (right) and comparing with the reference SCM (without nudging, *ke_T2*), the WRF simulation, the ERA Interim input data and the observations.**





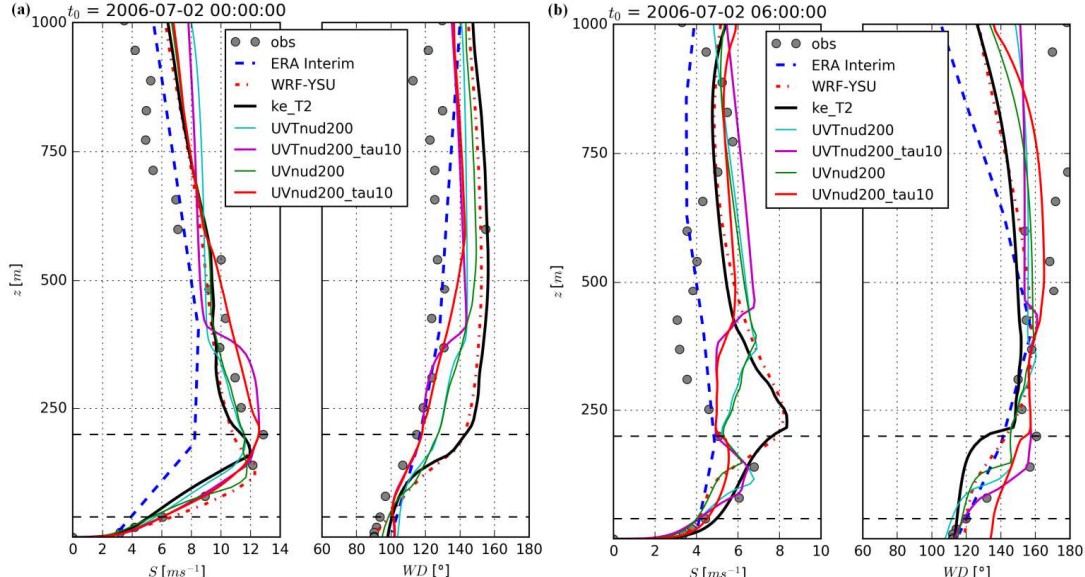

**Figure 13: Vertical profiles of horizontal wind speed _S_ and wind direction _WD_ at 2006-07-02 00:00:00 (a) and 06:00:00 (b) using different nudging strategies as per Table 1 and compared with the reference SCM (without nudging, _ke_T2_), the WRF simulation, the ERA Interim input data and the observations.**