# Peer review of "A methodology for the design and testing of atmospheric boundary layer models for wind energy applications"

_Wind Energy Science, 2016_

## Referee Comment (RC1) · Anonymous Referee #1 · 11 Oct 2016

Review of "Atmospheric boundary layer modeling based on mesoscale tendencies and data assimilation at microscale" by J. Sanz Rodrigo, M. Churchfield and B. Kosovic

The manuscript describes a series of simulations done with a RANS-type model with different closures and used to simulate the various GABLS intercomparison cases. The GABLS cases are setup in a single-column mode and the GABLS 1 and 2 tests show high consistency between the various closures. The GABLS 3 simulations are initialized and forced by advection terms derived from mesoscale modeling using the WRF model downscaling from ERA-interim reanalysis.

I believe the results are interesting, but the manuscript fails to argue why these experiments are relevant to wind energy applications. What do we really learn from such

a setup? In this rather flat and uniform site small-scale advection terms are probably unimportant and thus the WRF advective terms are closer to reality. But I fail to see how such a setup could be used in simulations in more complex terrain, where the small scale horizontal terms become more important. This needs to be addressed both in the introduction and then again in the discussion and conclusion section.

I have also a few more technical issues.

1. I believe the title is a bit misleading. What is done in this paper is not really data assimilation. There is a strong debate in the meteorological community, which does not consider "nudging" a data assimilation technique. Data assimilation methods take into account the error characteristics of the data being assimilated. Here that is not taken into account. I suggest that you substitute "data assimilation" by simply nudging or newtonian relaxation.

2. Is the setup double counting the forcing of the WRF data in the RANS model? Both advection terms and nudging are used to drive the results towards the results of the WRF simulations.

3. There are serious problems with the WRF setup. It is not appropriate to downscale directly from ERA-Interim at a grid spacing of ∼80 km to 9 km. The scales are just too different, and the simulation is likely missing some of the large-scale forcing. Please see http://www2.mmm.ucar.edu/wrf/users/workshops/WS2014/ppts/best_prac_wrf.pdf page on "Nesting, Resolution and Domain Size".

4. In P3. L9-10. I don't really understand what you mean by "... there is a strong coupling between the geostrophic wind speed and the surface temperature." Please explain.

5. In P12, L26. 'Interestingly, the advective wind makes a 360 deg turn throughout the cycle'. But the "large" fluctuations in direction advection coincide with very small fluctuations in wind speed advection. So, they are probably not meaningful. In this

case it would be better to avoid the separation in wind speed and direction and use zonal and meridional wind.

6. Figure 3. I don't understand what this figure is about. What does sensitivity analysis means? What are the figures of? What are the colors? The figure caption is not very informative.

Other minor editorial changes.

Many places with jargon or informal english. P2. L10-11, "... include relevant physics...", should be include the relevant physical processes, or physical-dynamical processes. "physics" does not mean anything P3, L29, "first-order physics" P17, L6, "are way off", L7 "doesn't", L17 "haven't".

Paragraph starting in P3, L20. I think the grammar is quite inconsistent. What is it that you mean?

P4, L2, "to count with" better will be "to have"

P9, L2. K h-1 is not standard units. K/hour would be better.

P9, L20, "are almost never happening" is weird. How about "almost never happen"?

P9, L25, "... based on observations, from the CASES-99..." the comma should not be there.

P11, L 20, the USGS land use is a map of surface characteristics parameter not a model.

P14, L19. There is rather a long jump in figure number. To Fig 12, then back to Fig 11. It is easier for the reader if these are in order in the text.

---

## Author Comment (AC1) · 11 Nov 2016

Thanks for your comments. Let us address them one by one following your numbering:

0.  About the general comment of "the manuscript fails to argue why these experiments are relevant to wind energy application". We believe that this is well motivated in the first two paragraphs of the introduction. Wind energy flow models in industry are still largely based on surface-layer modeling in neutral conditions while turbines and wind farms are getting very large and and "include relevant physics like Coriolis as well as realistic large-scale forcing and appropriate turbulent scaling, dependent on thermal stratification, from the surface layer to the free atmosphere". Hence, the needs for wind energy are clearly stated. When developing these models one has to make

sure that relevant physics are included and their impact quantified systematically. The GABLS series of benchmarks, developed by the meteorology community, is an excellent verification suite for the design of ABL wind energy models. Further motivation about meso-micro is provided in the review paper (Sanz Rodrigo et al., 2016).

1. About "data assimilation" or not. We don't agree that the method can't be called data assimilation. Nudging is a well established data assimilation technique, widely used by the meteorological community. Its simplicity shall make it also a popular method in wind energy where we typically count with site measurements. Anyway, we could agree that changing "data assimilation" by "nudging" will be more specific about the method and will avoid the controversy.

2. About double-counting. We demonstrate that adding advection has value to the predictions before using nudging. Nudging is used at microscale to correct the errors of the WRF-SCM simulation towards the observations. This is not double-counting, advection is a genuine atmospheric force while nudging is not.

3. About the WRF set-up not being appropriate. We don't agree that using a higher resolution domain will miss large scale forcings that a lower resolution domain would capture. On the contrary, if the time step still yields under-critical CFL numbers, the same forcings from the coarser domain will be better resolved with the higher resolution domain. The guidelines from NCAR suggest a resolution ratio of 3 to 5 when changing from one nest to the next as trade-off between the scales resolved in each domain and the computational cost. We use a higher resolution parent domain to have all the nests with the same grid size and use a larger number of cpus in the simulation. This is particularly important in WRF-LES simulations of a follow-up work.

4. About the strong coupling of geostrophic wind and surface temperature. This literal conclusion comes from the assessment of the GABLS2 case, where another diurnal cycle was under discussion. In GABLS2, the surface temperature was prescribed while in GABLS3 it was allowed to respond to the forcings as a result of the surface
model. Holtslag et al. (2007) showed the impact of prescribing the temperature or not and found significant differences in stable conditions. That's why, when designing GABLS3, they decided to allow coupling of forcings through the surface model instead of prescribing the surface temperature.

5. About using velocity magnitude and direction instead of U and V components. We agree that the large changes of advection direction coincide with low advection velocity magnitude. We'd rather use magnitude instead of components because we are talking about forcings at rotor level and, hence, it is more meaningfull to talk about rotor-based quantities of interest that are later use in the validation. Nevertheless, we'll change the test to: "Interestingly, the advective wind direction makes a 360° turn throughout the cycle, although at relatively small advection speed".

6. About Fig.3 meaning. We believe that the caption of the Figure, together with the explanations in the main text, is comprehensive enough to explaing what's going on. Based on the GABLS1 set-up that runs a uniformly cooled ABL to a quasi-steady state, we quantify some characteristics of the resulting profiles based on changes on the driving forcings (cooling rate and geostrophic wind). The colors show different stability classes. We don't add a legend because the contours already include labels.

We'll review the text to include the editorial changes and avoid jargon or informal english.

---

## Referee Comment (RC2) · Anonymous Referee #2 · 17 Nov 2016

This manuscript assess the possibility to drive micro-scale RANS models for wind energy application, with forcing's from meso-scale models. On the basis of the three GABLS cases it nicely analyze the performance when different model configurations and model couplings are done.

The authors clearly present the results of a quite complex exercise. The manuscript is clearly written and graphs are clear and to the point.

P2 L6 What "larger scales" are meant here temporal or spatial?

P2 L5 MOST is not the theory for neutral conditions, it is the theory that extends from neutral to non-neutral conditions.

[Figure]

P2 L10 At this stage it is unclear what "micro-scale models" are. P2L15 Do the authors mean "ABL models" or "microscale models"?

P4L3" Hence, contrary to the original GABLS3 set-up, we allow the mesoscale forcing to retain its uncertainties, for the sake of a more generalized mesoscale-to microscale methodology, and then relax the microscale model simulation towards the profile observations to correct the hour-to-hour bias." I think this needs some more wording to become clear to the general reader.

P4 L26 How is this coordinate system oriented?

P4 L23 "This meso-micro methodology" Do the authors mean a "one –way coupling" ? Please reformulate.

P5 L 8-9. Why the subscript "pbl" for the turbulent diffusion tendency

P5 L17 are these terms height dependent? Unclear how the height information of the observations is incorporated.

P6 L1 or is it just the diurnal time scale itself?

P9 Conclusion for GABLS1 not much difference Fig 3 needs some more explanation. Stability is plotted with a color code

P10L L4 humidity is not relevant as long as clouds are absent.

P10 L9-12: This is unclear formulated. Are the 5 cycles 5 times the 48 hour periods? How then can consecutive days have almost the same temperature and wind? Also look at the formulation of the caption of fig 4

P10 L17-20. A higher k can be a sign of the model being less dissipative, as it is unable to get rid of the turbulent kinetic energy.

P11L6 Is 19 m/s correct?

P11 L10-13 Is there anything to say about the quality of the advective terms in the

meso-scale simulation?

P12 L3 Fig 7. If I add the individual components I would expect the signature of U_cor to stand out more clearly in U_tend.

P12 L3 Fig 7. Bosveld et al. (2014) attributed the strong tendency after midnight to U_adv. Please comment.

P12 L3 Fig 7. The strong peak at midnight in U-adv is after 60 minute filtering only 60 minute wide. This means that in the original data it is even narrower in time, and may indicate a very sharp front. Much sharper then is found in the RACMO run of Bosveld et al. (2014) and much sharper then observed.

Textual comments:

P1L11 insert " cases" P1L15 " from the Cabauw meteorological tower" P2 L5 "site measurements at standard height" P2 L5 "relying" P3 L13 Bass -> Baas (see also P4 L18 and P4 L22) P4 L1 unclear sentence P9 L3 long -> high P9 L9 than -> as P12 L21 "even though the filtering process, .." incorrect formulation P12 L25 "a" should be "at", "than" should be "as". In general please check carefully throughout the manuscript for misspelling! P12 L30 add "This results in an imbalance of forces ….." P15 L1 the term "footprint" is confusing, you may want to use " structure" P15 L1 change to " . . . even though more simplified physics is used."
* * *

---

## Referee Comment (RC3) · B. Holtslag (Referee) · 22 Nov 2016

This paper uses the first 3 GEWEX Atmospheric Boundary-Layer Studies (GABLS) to develop a methodology for the design and testing of (some) atmospheric boundary-layer (ABL) models for wind energy applications. Overall, the authors did a good job in exploiting the GABLS results for model evaluation and they do provide additional and relevant information in addition to what is already covered in the cited GABLS papers, in particular because of their focus on wind energy. I propose to accept it after some minor changes: 1. To better advertise the methodology I suggest to update the title into: "A methodology for the design and testing of boundary layer models for wind energy applications". 2. Systematically the name of Baas is misspelled in the paper,

so use Baas et al (not Bass et al). 3. Page 5, Eq2. I think it is confusing to use U and V also for the various tendencies of the wind components. The subscript does not really help, use another symbol (or drop Eq 2 completely, it has not much additional value anyway compared to Eq 1). 4. Similarly, in the meteorology journals "e" is used for TKE (not k, which is mostly used for Von Karman constant). 5. Page 8, line 5. Strictly speaking this is not MOS but a result of using the well-established log profile for neutral conditions. 6. Page 9, line 13: Please refer to the original paper for Charnock, not your own earlier application unless you made an important extension. 7. Page 9, I like the idea of using off-shore site conditions to test the models for GABLS1 but the cooling tendencies used do not seem to be very realistic for ocean conditions. 8. Page 12, the authors in referring to Blackadar (1957) may also have a look at the more recent extensions by vdWiel et al (JAS, 2010) and Baas et al (QJ, 2012). 9. Page 13, I do not understand why and how the power law is needed to discuss wind shear results. Please explain. 10. Figures are very difficult to read (at least for my senior eyes), so please magnify or enhance otherwise. 11. Regarding Fig 2, it would also be instructive to show the hodographs (see Svensson and Holtslag, 2009) 12. I suggest no to use acronyms as Qol (it does not read well).

Good luck with the final version!

Bert Holtslag, 19 Nov 2016

---

## Author Comment (AC3) · 28 Nov 2016

Thanks for the review. Let's go point by point:

1."A methodology for the design and testing of boundary layer models for wind energy applications" Yes, since GABLS 1 and 2 are also added to the paper, I guess the "methodology" is more relevant than the "model". We'll change the title to this one.

2. I guess the typo got carried away...

3. Equation (2) is introduced to simplify Eq (1) and use the notation to introduce the mesoscale forcings in equation (3) when we switch from mesoscale to microscale modeling. We use velocity components U and V because the units are in terms of m/s. We

also use this equation and notation in connection to Figures 7 and 8. We'd rather keep the Equation and add more information about the notation used, in particular to "pbl", as pointed out by reviewer #2.

4. E or k... We adopted here the wind engineering notation that normally uses k instead of E for the turbulent kinetic energy. Then, we speak about k-epsilon models, etc. For the von Karman constant we use kappa, the Greek letter, to differentiate.

5. We will replace this sentence with: " ... a relationship amongst k-epsilon coefficients is prescribed in order to obtain consistency with well-established log profiles in surface-layer neutral conditions (Richards and Hoxey, 1993):"

6. Charnock relation. Yes, I will add the original reference to Charnock. In Sanz Rodrigo (2011) the Charnock coefficient is calibrated using Fino1 measurements.

7. Well, the range of cooling rates is not necessarily limited to offshore conditions. It is just an exercise using the latitude and roughness length of Fino1 and then using a wide range of cooling rates leading to z/L values from -20 to 20 at 70 m. In Sanz Rodrigo et al. (2015) we analyzed flux-profile relationships that showed stabilities mainly in the range from -2 to 2 at 80 m. We'll add this reference to indicate the relevant range of stabilities at Fino1.

8. Agreed, we just wanted to provide a historical reference. We'll add the more recent references as well.

9. The power-law exponent "alpha" is convention in wind energy to quantify the wind shear. Similarly, a linear fit is used to characterize wind direction veer. We used these engineering methods here following the wind energy convention but we also mention that their suitability needs to be checked: "While these fitting functions are commonly used in wind energy, their suitability in LLJ conditions is questionable. The regression coefficient from the fitting can be used to determine this suitability."

10. Figures 12 and 13 are efficient ways of plotting a large number of simulations such

that they can be compared. They could be magnified (like Figure 11) but then they should be separated into 4 figures. We believe that it is easier to compare the results with the compact solution of the paper but we can adopt the solution of 4 figures if the editor also prefers larger figures.

11. Hodographs... Maybe this is another meteorology vs wind-energy convention. In wind energy we tend to use wind energy and wind speed rather than velocity components. The paper already includes 13 Figures, which might be 15 following the previous point. We'd rather not include more figures.

12. Agreed, let's not use the QoI acronym.

―――――――――――――――――

---

## Author Response (AR1)

**Atmospheric boundary layer modeling based on mesoscale tendencies and data assimilation at microscale**

Final review, 22 December 2016

Thanks for the online discussion. Now let's address all the reviewer's comments in orderly fashion.  We'll try to make the answers self-contained.

**Answer to Reviewer #1**

*0. The manuscript fails to argue why these experiments are relevant to wind energy application. What do we really learn from such a setup? In this rather flat and uniform site small-scale advection terms are probably unimportant and thus the WRF advective terms are closer to reality. But I fail to see how such a setup could be used in simulations in more complex terrain, where the small scale horizontal terms become more important. This needs to be addressed both in the introduction and then again in the discussion and conclusion section.*

The following text has been added to the manuscript to extend the discussion about how this methodology is relevant for a future improved 'wind-energy' model chain.

At the introduction:

"Sanz Rodrigo et al. (2016) reviews the state-of-the-art of wind farm flow modeling and methodologies and challenges for mesoscale-to-microscale coupling."

At the end of the paper, in the discussion and conclusions section:

" SCM simulations over horizontally homogeneous terrain is a convenient methodology for the design of ABL models given its simpler code implementation and interpretation of results compared to a 3D setting in heterogeneous conditions. This allows to test surface boundary conditions, turbulence models and large-scale forcings more efficiently before implementing them in a 3D microscale model. In 3D, advection would be solved by the model through surface heterogeneities and velocity gradients across the lateral boundaries. Spatial-averaged, height and time dependent mesoscale forcing from horizontal pressure gradients could be introduced as a column body force throughout the 3D domain similarly as it has been done in GABLS3. By spatial averaging over a larger scale than the microscale domain, we expect to filter out disturbances in the pressure gradient due to unresolved topography in the mesoscale model. These topographic effects will be modelled with a high-resolution topographic model in the 3D microscale simulation. Such model-chain would still assume that the mesoscale forcing is horizontally homogeneous throughout the microscale domain but changes with height and time through source terms in the momentum equations. Nudging local corrections would be introduced through horizontal and vertical weight functions that limit the correction to the local vicinity of the observation sites as it is done in mesoscale models (Stauffer and Seaman, 1990). This relatively simple implementation of meso-micro coupling is valid for RANS and LES models and allows easier characterization of mesoscale inputs than using 3D fields."

*1. I believe the title is a bit misleading. What is done in this paper is not really data assimilation. There is a strong debate in the meteorological community, which does not consider "nudging" a data assimilation technique. Data assimilation methods take into account the error characteristics of the data being assimilated. Here that is not taken into account. I suggest that you substitute "data assimilation" by simply nudging or newtonian relaxation.*

Following this and the third reviewer comments about the title we have accepted the title proposed by the third reviewer: "**A methodology for the design and testing of atmospheric boundary layer models for wind energy applications**"

We also replaced "data assimilation" by "nudging" or "bias-correction" in the text.

**2. Is the setup double counting the forcing of the WRF data in the RANS model? Both advection terms and nudging are used to drive the results towards the results of the WRF simulations.**

We demonstrate that adding advection has value to the predictions before using nudging. Nudging is used at microscale to correct the errors of the WRF-SCM simulation towards the observations. This is not double-counting, advection is a genuine atmospheric force while nudging is not.

**3. There are serious problems with the WRF setup. It is not appropriate to downscale directly from ERA-Interim at a grid spacing of ~80 km to 9 km. The scales are just too different, and the simulation is likely missing some of the large-scale forcing. Please see http://www2.mmm.ucar.edu/wrf/users/workshops/WS2014/ppts/best_prac_wrf.pdf page on "Nesting, Resolution and Domain Size".**

We don't agree that using a higher resolution domain will miss large scale forcings that a lower resolution domain would capture. On the contrary, if the time step still yields under-critical CFL numbers, the same forcings from the coarser domain will be better resolved with the higher resolution domain. The guidelines from NCAR suggest a resolution ratio of 3 to 5 when changing from one nest to the next as trade-off between the scales resolved in each domain and the computational cost. We use a higher resolution parent domain to have all the nests with the same grid size and use a larger number of cpus in the simulation. This is particularly important in WRF-LES simulations of a follow-up work.

**4. In P3. L9-10. I don't really understand what you mean by ". . . there is a strong coupling between the geostrophic wind speed and the surface temperature." Please explain.**

This literal conclusion comes from the assessment of the GABLS2 case in Holtslag et al. (2007), where another diurnal cycle was under discussion. In GABLS2, the surface temperature was prescribed while in GABLS3 it was allowed to respond to the forcings as a result of the surface model. Holtslag et al. (2007) showed the impact of prescribing the temperature or not and found significant differences in stable conditions. That's why, when designing GABLS3, they decided to allow coupling of forcings through the surface model instead of prescribing the surface temperature.

**5. In P12, L26. 'Interestingly, the advective wind makes a 360 deg turn throughout the cycle'. But the "large" fluctuations in direction advection coincide with very small fluctuations in wind speed advection. So, they are probably not meaningful. In this case it would be better to avoid the separation in wind speed and direction and use zonal and meridional wind.**

We agree that the large changes of advection direction coincide with low advection velocity magnitude. We'd rather use magnitude instead of components because we are talking about forcings at rotor level and, hence, it is more meaningfull to talk about rotor-based quantities of interest that are later use in the validation. Nevertheless, we'll change the test to:
"Interestingly, the advective wind direction makes a 360◦ turn throughout the cycle, although at relatively small advection speed"

**6. I don't understand what this figure is about. What does sensitivity analysis means? What are the figures of? What are the colors? The figure caption is not very informative.**

We believe that the caption of the Figure, together with the explanations in the main text, is comprehensive enough to explain what's going on. Based on the GABLS1 set-up that runs a

uniformly cooled ABL to a quasi-steady state, we quantify some characteristics of the resulting profiles based on changes on the driving forcings (cooling rate and geostrophic wind). The colors show different stability classes.

More information about the color scale and stability classes is provided now in the caption:

"Stability levels according to Sanz Rodrigo et al. (2014): near neutral (white): $0 < \zeta < 0.02$; weakly stable: $0.02 < \zeta < 0.2$; stable $0.2 < \zeta < 0.6$; very stable $0.6 < \zeta < 2$; extremely stable $\zeta > 2$ (symmetric range in unstable conditions in red) "

***Minor editorial changes:***

***Many places with jargon or informal english. P2. L10-11, ". . . include relevant physics. . .", should be include the relevant physical processes, or physical-dynamical processes. "physics" does not mean anything P3, L29, "first-order physics" P17, L6, "are way off", L7 "doesn't", L17 "haven't". Paragraph starting in P3, L20. I think the grammar is quite inconsistent. What is it that you mean? P4, L2, "to count with" better will be "to have" P9, L2. K h-1 is not standard units. K/hour would be better. P9, L20, "are almost never happening" is weird. How about "almost never happen"? P9, L25, ". . . based on observations, from the CASES-99. . ." the comma should not be there. P11, L 20, the USGS land use is a map of surface characteristics parameter not a model. P14, L19. There is rather a long jump in figure number. To Fig 12, then back to Fig 11. It is easier for the reader if these are in order in the text.***

We have reviewed the text to include the editorial changes and avoid jargon or informal English.

**Answer to Reviewer #2**

**P2 L6 What "larger scales" are meant here temporal or spatial?**

Here "large-scale" is anything larger than microscale in a broad sense.

"(than microscale)" added to text:

" At larger scales (than microscale), the long-term wind climatology is typically determined from a combination of historical measurements and simulations from mesoscale meteorological models at a horizontal resolution of a few kilometers."

**P2 L5 MOST is not the theory for neutral conditions, it is the theory that extends from neutral to non-neutral conditions.**

Yes, you are right. I meant MOST applied in neutral conditions. Text changed to:

"have been traditionally based on site measurements and microscale flow models relaying on Monin-Obukhov surface-layer theory (MOST, Monin and Obukhov, 1954) that assume steady-state and are typically applied in neutral atmospheric conditions"

**P2 L10 At this stage it is unclear what "micro-scale models" are.**

"Microscale" is simply defined in the first paragraph as the flow around and within a wind farm. We believe this is how the wind energy community understand microscale models.

**P2L15 Do the authors mean "ABL models" or "microscale models"?**

We'd rather use ABL models as the backbone of microscale models dealing with atmospheric boundary-layer turbulence. The paper is about development of ABL models in flat terrain, not microscale models that would include other complexities (terrain, wind farms, etc).

**P4L3" Hence, contrary to the original GABLS3 set-up, we allow the mesoscale forcing to retain its uncertainties, for the sake of a more generalized mesoscale-to microscale methodology, and then relax the microscale model simulation towards the profile observations to correct the hour-to-hour bias." I think this needs some more wording to become clear to the general reader.**

How about:

"Hence, contrary to the original GABLS3 set-up, for the sake of a more generalized mesoscale-to microscale methodology, we propose using the large-scale tendencies computed by a mesoscale model as driving forces at microscale without introducing any correction based on measurements. Then, at microscale, the simulation can be dynamically relaxed to the profile observations to correct the hour-to-hour bias."

**P4 L26 How is this coordinate system oriented?**

Yes, we should say

"natural Cartesian coordinates (x --> East, y--> North, z --> vertical)"

**P4 L23 "This meso-micro methodology" Do the authors mean a "one –way coupling" ? Please reformulate.**

Yes, it is one-way coupling. We have added this distinction.

**P5 L 8-9. Why the subscript "pbl" for the turbulent diffusion tendency**

"pbl" for planetary-boundary layer. We use this term here following the same term in the WRF community to relate to boundary-layer parameterizations or "PBL schemes". We've added this explanation as follows:

"...U_pbl and V_pbl are the turbulent diffusion wind components (equivalent to the so-called planetary-boundary layer (PBL) scheme in mesoscale models)".

**P5 L17 are these terms height dependent? Unclear how the height information of the observations is incorporated.**

Yes, they are all height and time dependent. Observations are nudged according to (4) by using a height-dependent weight function w_z as described in the text.

**P6 L1 or is it just the diurnal time scale itself?**

The time-scale tau_nud simply determines how gradual is the bias-correction introduced

**P9 Conclusion for GABLS1 not much difference Fig 3 needs some more explanation. Stability is plotted with a color code**

Fig.3 contour plots summarize the profile characteristics, using a range of surface cooling rates and geostrophic wind magnitudes, after 9-hr of GABLS1-like simulations to a quasi-steady state. More information about the color scale and stability classes is provided in the caption:

"Stability levels according to Sanz Rodrigo et al. (2014): near neutral (white): $0 < \zeta < 0.02$; weakly stable: $0.02 < \zeta < 0.2$; stable $0.2 < \zeta < 0.6$; very stable $0.6 < \zeta < 2$; extremely stable $\zeta > 2$ (symmetric range in unstable conditions in red) "

**P10L L4 humidity is not relevant as long as clouds are absent.**

Yes, this is true, although wind energy "microscale" models typically do not include the humidity equation.

**P10 L9-12: This is unclear formulated. Are the 5 cycles 5 times the 48 hour periods? How then can consecutive days have almost the same temperature and wind? Also look at the formulation of the caption of fig 4**

Yes, the "cycle" here corresponds to the 48 hour long period of Fig.4, which is repeated 5 times to obtain equilibrium (difference between cycle 5 and cycle 4 is small). Fig. 4 dashed line shows two of these cycles. We have removed the time labels 25-Oct-1999 and 26-Oct-1999 since the simulation time after the first cycle does not correspond to real time.

**P10 L17-20. A higher k can be a sign of the model being less dissipative, as it is unable to get rid of the turbulent kinetic energy.**

Yes, you could say that. We'll simply say:

"As the closure order is increased, higher turbulent kinetic energy is observed. Higher mixing..."

**P11L6 Is 19 m/s correct?**

Typo, it should say 5 to 10 m/s

**P11 L10-13 Is there anything to say about the quality of the advective terms in the meso-scale simulation?**

The assessment of WRF from Kleczek et al (2014) doesn't include advective terms.

**P12 L3 Fig 7. If I add the individual components I would expect the signature of U_cor to stand out more clearly in U_tend.**

In the text, we mention "curvature" tendencies not being significant. These are not the Coriolis forces but terms that appear in WRF due to having a curvilinear coordinate system. To avoid confusion we will say:

"Curvature, due to curvilinear coordinate system in WRF, and horizontal..."

**P12 L3 Fig 7. Bosveld et al. (2014) attributed the strong tendency after midnight to U_adv. Please comment.**

Yes, there is a clear signature of U_adv in the momentum tendencies during the night. While the timing and magnitude of advection tendencies is difficult to predict, the results of the sensitivity analysis showed that not including this forcing resulted in worse results than including it. This is discussed in the text and was also mentioned by Bosveld et al. (2104)

**P12 L3 Fig 7. The strong peak at midnight in U-adv is after 60 minute filtering only 60 minute wide. This means that in the original data it is even narrower in time, and may indicate a very sharp front. Much sharper then is found in the RACMO run of Bosveld et al. (2014) and much sharper then observed.**

Yes, the advection tendencies in Bosveld et al. (2014) show a broader peak at midnight. It is difficult to say where the differences come from since RACMO simulations in Bosveld et al. and our simulations with WRF were done at different resolutions and with different input data.

We add this point in the text:

"Advection tendencies show narrower peaks compared to those from Bosveld et al. (2014a). It is difficult to say where these differences are coming from since we used different input data and horizontal and temporal resolutions."

**Textual comments:**

**P1L11 insert " cases" P1L15 " from the Cabauw meteorological tower" P2 L5 "site measurements at standard height" P2 L5 "relying" P3 L13 Bass -> Baas (see also P4 L18 and P4 L22) P4 L1 unclear sentence P9 L3 long -> high P9 L9 than -> as P12 L21 "even though the filtering process, .." incorrect formulation P12 L25 "a" should be "at", "than" should be "as". In general please check carefully throughout the manuscript for misspelling! P12 L30 add "This results in an imbalance of forces . . ..." P15 L1 the term "footprint" is confusing, you may want to use " structure" P15 L1 change to " . . . even though more simplified physics is used."**

Thanks for the editorial changes. We have considered them in the revised version.

**Comments to Reviewer #3**

*1. To better advertise the methodology I suggest to update the title into* "**A methodology for the design and testing of boundary layer models for wind energy applications**"

Yes, since GABLS 1 and 2 are also added to the paper, I guess the "methodology" is more relevant than the "model". We shall change the title to:

**"A methodology for the design and testing of atmospheric boundary layer models for wind energy applications"**

*2. Systematically the name of Baas is misspelled in the paper*

I guess the typo got carried away...

*3. Page 5, Eq2. I think it is confusing to use U and V also for the various tendencies of the wind components. The subscript does not really help, use another symbol (or drop Eq 2 completely, it has not much additional value anyway compared to Eq 1)*

Equation (2) is introduced to simplify Eq (1) and use the notation to introduce the mesoscale forcings in equation (3) when we switch from mesoscale to microscale modeling. We use velocity components U and V because the units are in terms of m/s. We also use this equation and notation in connection to Figures 7 and 8. We'd rather keep the Equation and add more information about the notation used, in particular to "pbl", as pointed out by reviewer #2.

*4. Similarly, in the meteorology journals "e" is used for TKE (not k, which is mostly used for Von Karman constant)*

We adopted here the wind engineering notation that normally uses k instead of E for the turbulent kinetic energy. Then, we speak about k-epsilon models, etc. For the von Karman constant we use kappa, the Greek letter, to differentiate.

*5. Page 8, line 5. Strictly speaking this is not MOS but a result of using the well-established log profile for neutral conditions.*

We have replaced this sentence with:

" ... a relationship amongst k-epsilon coefficients is prescribed to obtain consistency with well-established log profiles in surface-layer neutral conditions (Richards and Hoxey, 1993):"

*6. Page 9, line 13: Please refer to the original paper for Charnock, not your own earlier application unless you made an important extension*

Yes, we have added the original reference to Charnock. In Sanz Rodrigo (2011) the Charnock coefficient is calibrated using Fino1 measurements.

"friction velocity through the Charnock relation (Charnock 1955), calibrated for Fino-1 conditions in Sanz Rodrigo (2011), with $z_0$ = 0.0002 m being a representative value"

*7. Page 9, I like the idea of using off-shore site conditions to test the models for GABLS1 but the cooling tendencies used do not seem to be very realistic for ocean conditions*

Well, the range of cooling rates is not necessarily limited to offshore conditions. It is just an exercise using the latitude and roughness length of Fino1 and then using a wide range of cooling rates leading to z/L values from -20 to 20 at 70 m. In Sanz Rodrigo et al. (2015) we analyzed flux-profile relationships that showed stabilities mainly in the range from -2 to 2 at 80 m. We'll add this reference to indicate the relevant range of stabilities at Fino1.

"The stability parameter *z/L* at the reference height is also plotted following the stability classes defined in Sanz Rodrigo et al. (2014), where sonic measurements of the at Fino-1 show a stability range at 80 m from -2 to 2."

**8. Page 12, the authors in referring to Blackadar (1957) may also have a look at the more recent extensions by vdWiel et al (JAS, 2010) and Baas et al (QJ, 2012)**

Agreed, we just wanted to provide a historical reference. We've added the more recent references as well.

**9. Page 13, I do not understand why and how the power law is needed to discuss wind shear results. Please explain.**

The power-law exponent "alpha" is convention in wind energy to quantify the wind shear. Similarly, a linear fit is used to characterize wind direction veer. We used these engineering methods here following the wind energy convention but we also mention that their suitability needs to be checked: "While these fitting functions are commonly used in wind energy, their suitability in LLJ conditions is questionable. The regression coefficient from the fitting can be used to determine this suitability."

**10. Figures are very difficult to read (at least for my senior eyes), so please magnify or enhance otherwise.**

Figures 12 and 13 are efficient ways of plotting a large number of simulations such that they can be compared. They could be magnified (like Figure 11) but then they should be separated into 4 figures. We believe that it is easier to compare the results with the compact solution of the paper but we can adopt the solution of 4 figures if the editor also prefers larger figures.

**11. Regarding Fig 2, it would also be instructive to show the hodographs (see Svensson and Holtslag, 2009)**

Maybe this is another meteorology vs wind-energy convention. In wind energy we tend to use wind energy and wind speed rather than velocity components. The paper already includes 13 Figures, which might be 15 following the previous point. We'd rather not include more figures.

**12. I suggest no to use acronyms as QoI (it does not read well)**

Agreed, let's not use the QoI acronym.